# Select to Perfect: Imitating desired behavior from large multi-agent data

**Tim Franzmeyer**[*]    **Edith Elkind**    **Philip Torr**    **Jakob Foerster**[†]    **João F. Henriques**[†]

University of Oxford

## Abstract

AI agents are commonly trained with large datasets of demonstrations of human behavior. However, not all behaviors are equally safe or desirable. Desired characteristics for an AI agent can be expressed by assigning desirability scores, which we assume are not assigned to individual behaviors but to collective trajectories. For example, in a dataset of vehicle interactions, these scores might relate to the number of incidents that occurred. We first assess the effect of each individual agent's behavior on the collective desirability score, e.g., assessing how likely an agent is to cause incidents. This allows us to selectively imitate agents with a positive effect, e.g., only imitating agents that are unlikely to cause incidents. To enable this, we propose the concept of an agent's *Exchange Value*, which quantifies an individual agent's contribution to the collective desirability score. The Exchange Value is the expected change in desirability score when substituting the agent for a randomly selected agent. We propose additional methods for estimating Exchange Values from real-world datasets, enabling us to learn desired imitation policies that outperform relevant baselines. The project website can be found at `https://tinyurl.com/select-to-perfect`.

## 1 Introduction

Imitating human behaviors from large datasets is a promising technique for achieving human-AI and AI-AI interactions in complex environments (Carroll et al., 2019; , FAIR; He et al., 2023; Shih et al., 2022). However, such large datasets can contain undesirable human behaviors, making direct imitation problematic. Rather than imitating all behaviors, it may be preferable to ensure that AI agents imitate behaviors that align with predefined desirable characteristics. In this work, we assume that desirable characteristics are quantified as desirability scores given for each trajectory in the dataset. This is commonly the case when the evaluation of the desirability of individual actions is impractical or too expensive (Stiennon et al., 2020). Assigning desirability scores to collective trajectories may be the only viable option for complex datasets that involve multiple interacting agents. For instance, determining individual player contributions in a football match is difficult, while the final score is a readily-available measure of team performance.

We develop an imitation learning method for multi-agent datasets that ensures alignment with desirable characteristics – expressed through a Desired Value Function[1] (DVF) that assigns a score to each *collective* trajectory. This scenario is applicable to several areas that involve learning behavior from data of human groups. One example is a dataset of vehicle interactions, desirability scores indicating the number of incidents in a collective trajectory, and the aim to imitate only behavior that is unlikely to result in incidents (e.g., aiming to imitate driving with foresight). Similarly – given a dataset of social media conversation threads and desirability scores that indicate whether a thread has gone awry – one may want to only imitate behavior that reduces the chance of conversations going awry Chang & Danescu-Niculescu-Mizil (2019).

---

[*]frtim@robots.ox.ac.uk        [†]equal supervision

[1]The DVF itself is not sufficient to describe desired behavior completely, as it possibly only covers a subset of behavior, e.g., safety-relevant aspects. It is complementary to the more complex and nuanced behaviors that are obtained by imitating human demonstrations, providing guardrails or additional guidance.

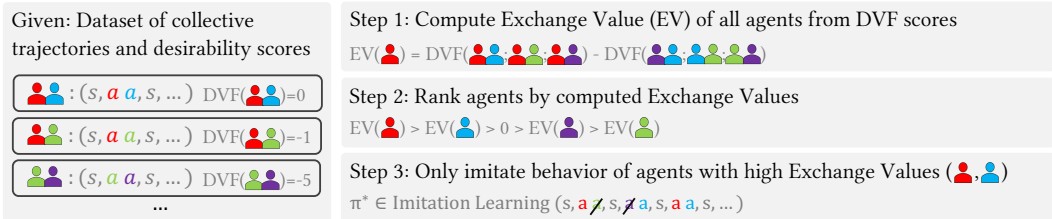

Figure 1: We are given a dataset composed of multi-agent trajectories generated by many individual agents, e.g., a dataset of cars driving in urban environments. In addition, the Desired Value Function (DVF) indicates the desirability score of a collective trajectory, e.g., the number of incidents that occurred. We first compute the Exchange Value (EV) of each agent, where a positive EV indicates that an agent increases the desirability score (e.g. an agent driving safely). We reformulate imitation learning to take into account the computed EVs, and achieve an imitation policy that is aligned with the DVF (e.g. only imitating the behavior of safe drivers).

Assessing the desirability of an individual agent's behavior involves gauging its impact on the *collective* desirability score. For instance, it requires evaluating whether a driver's behavior increases the likelihood of accidents, or whether a user's behavior increases the likelihood of a conversation going awry. This is termed the *credit assignment problem* (Shapley, 1953), akin to fairly dividing the value produced by a group of players among the players themselves. The credit assignment problem proves complex in real-world scenarios due to three main factors (see Figure 2 for details): First, many scenarios only permit specific group sizes.This makes Shapley Values (Shapley, 1953) – a concept commonly used in Economics for credit assignment – inapplicable, as it relies on the comparisons of groups of different sizes (e.g., Shapley Values are not applicable to football players, as football is a game of 11 players and a group of 12 is never observed.) Second, real-world datasets for large groups are almost always incomplete, i.e., they do not contain trajectories for all (combinatorially many) possible groups of agents. Third, datasets of human interactions may be *fully anonymized* by assigning one-time-use IDs. In this case, if an agent is present in two trajectories, it will appear in the dataset as if it is *two different agents*, making the credit assignment problem degenerate. This requires incorporating individual behavior information in addition to the information about collective outcomes.

To address these challenges, we propose Exchange Values (EVs), akin to Shapley Values, which quantify an agent's contribution as the expected change in desirability when substituting the agent randomly. The EV of an agent can be understood as the expected change in value when substituting the agent with another randomly selected agent – or as comparing the average value of all groups that include the agent to that of all groups not including the agent (see Step 1 in Figure 1). EVs are applicable to scenarios with fixed group sizes, making them more versatile. We introduce EV-Clustering that estimates EVs from incomplete datasets by maximizing inter-cluster variance. We show a theoretical connection to clustering by *unobserved* individual contributions and adapt this method to fully-anonymized datasets, by considering low-level behavioral information.

We introduce Exchange Value based Behavior Cloning (*EV2BC*), which imitates large datasets by only imitating the behavior of selected agents with EVs higher than a tuneable threshold (see Figure 1). This approach allows learning from interactions with agents with all behaviors, without necessarily imitating them. This is not possible when simply excluding all trajectories with a low collective desirability score, i.e., selectively imitating based on collective scores instead of individual contributions. We find that EV2BC outperforms standard behavior cloning, offline RL, and selective imitation based on collective scores in challenging environments, such as the StarCraft Multi-Agent Challenge (Samvelyan et al., 2019). Our work makes the following contributions:

- We introduce *Exchange Values* (Def. 4.1) to compute an agent's individual contribution to a collective value function and show their relation to Shapley Values.
- We propose *EV-Clustering* (Def. 4.4) to estimate contributions from incomplete datasets and show a theoretical connection to clustering agents by their unobserved individual contributions.
- We empirically demonstrate how EVs can be estimated from fully-anonymized data and employ *EV2BC* (Def. 4.5) to learn policies aligned with the DVF, outperforming relevant baselines.

| Example Scenario | Dataset characteristics | | | Example observation dataset of agents 🧑, 🧑, 🧑, ... | Shapley Values Applicable | Exchange Values Applicable | Exchange Value Computation |
|---|---|---|---|---|---|---|---|
| | All Group Sizes | All Observations | Known Agent Identities | | | | |
| **Ideal**
All group sizes are permitted.
All possible combinations of agents (all groups) are observed. | ✓ | ✓ | ✓ | DVF(🧑) DVF(🧑) DVF(🧑) ...
DVF(🧑🧑) DVF(🧑🧑) DVF(🧑🧑) ...
DVF(🧑🧑🧑) ...
... | ✓ | ✓ | Exact |
| **Group-Limited**
Only specific group sizes are permitted.
A football game has 11 players. | ✗ | ✓ | ✓ | DVF(🧑🧑) DVF(🧑🧑) DVF(🧑🧑) ... | ✗ | ✓ | Exact |
| **Low-Data**
Not all permitted groups are observed.
Two football players might never play for the same team. | ✗ | ✗ | ✓ | DVF(🧑🧑) DVF(🧑🧑) ... | ✗ | ✓ | Estimated (potentially with EV-Clustering) |
| **Degenerate**
Anonymized with one-time-use IDs.
New ID for a player in each game played. | ✗ | ✗ | ✗ | DVF(🧑🧑) DVF(🧑🧑) ...
Behaviour clustering
DVF(🔴🔴) DVF(🔵🟢) ... | ✗ | ✓ | Estimated with EV-Clustering and behaviour information from $\tau$ |

Figure 2: Overview of different characteristics of real-world datasets, and whether Shapley Values and Exchange Values (EVs) are applicable to compute contributions of individual agents to the DVF.

## 2 RELATED WORK

Most previous work on aligning AI agents' policies with desired value functions either relies on simple hand-crafted rules (Xu et al., 2020; , FAIR), which do not scale to complex environments, or performs postprocessing of imitation policies with fine-tuning (Stiennon et al., 2020; Ouyang et al., 2022; Glaese et al., 2022; Bai et al., 2022), which requires access to the environment or a simulator. In language modeling, Korbak et al. (2023) showed that accounting for the alignment of behavior with the DVF already during imitation learning yields results superior to fine-tuning after-the-fact, however, their approach considers an agent-specific value function. In contrast, we consider learning a policy aligned with a collective value function, and from offline data alone.

Credit assignment in multi-agent systems was initially studied in Economics (Shapley, 1953). Subsequently, Shapley Values (Shapley, 1953) and related concepts have been applied in multi-agent reinforcement learning to distribute rewards among individual agents during the learning process (Chang et al., 2003; Foerster et al., 2018; Nguyen et al., 2018; Wang et al., 2020; Li et al., 2021; Wang et al., 2022). Outside of policy learning, Heuillet et al. (2022) used Shapley Values to analyze agent contributions in multi-agent environments, however this requires privileged access to a simulator, in order to replace agents with randomly-acting agents. In contrast to Shapley Values, the applicability of EVs to all group sizes allows us to omit the need to simulate infeasible coalitions.

In contrast to this work, existing work in multi-agent imitation learning typically assumes observations to be generated by optimal agents, as well as simulator access (Le et al., 2017; Song et al., 2018; Yu et al., 2019). Similar to our framework, offline multi-agent reinforcement learning (Jiang & Lu, 2021; Tseng et al., 2022; Tian et al., 2022) involves policy learning from multi-agent demonstrations using offline data alone, however, it assumes a dense reward signal to be given, while the DVF assigns a single score per collective trajectory.

In single-agent settings, a large body of work investigates estimating demonstrator expertise to enhance imitation learning (Chen et al., 2021; Zhang et al., 2021; Cao & Sadigh, 2021; Sasaki & Yamashina, 2021; Beliaev et al., 2022; Yang et al., 2021). However, these methods do not translate to the multi-agent setting due to the challenge of credit assignment.

To the best of our knowledge, no prior work has considered the problem of imitating multi-agent datasets containing mixed behaviors, while ensuring alignment with a collective value function.

## 3 BACKGROUND AND NOTATION

**Markov Game.** We consider Markov Games (Littman, 1994), which generalize Markov Decision Processes (MDPs) to multi-agent scenarios. In a Markov Game, agents interact in a common environment. At time step $t$, each agent (the $i$th of a total of $m$ agents) takes the action $a_i^t$ and the environ-

ment transitions from state $s^t$ to $s^{t+1}$. A reduced Markov game (without rewards) is then defined by a state space $\mathcal{S}$ ($s^t \in \mathcal{S}$), a distribution of initial states $\eta$, the action space $\mathcal{A}_i$ ($a_i^t \in \mathcal{A}_i$) of each agent $i$, an environment state transition probability $P(s^{t+1}|s^t, a_1, \ldots, a_m)$ and the episode length $T$. We denote this Markov Game as $\mathcal{M} = (\mathcal{S}, \mathcal{A}, P, T)$, with collective trajectories $\tau = (s_0, \mathbf{a}_0, \ldots, s_T)$.

**Set of multi-agent demonstrations generated by many agents.** We consider a Markov game $\mathcal{M}$ of $m$ agents and a set of demonstrator agents $N = \{1, ..., n\}$ where $n \geq m$. The Markov Game is further assumed to be symmetric (we can change the ordering of players without changing the game). The demonstration set $\mathcal{D}$ captures interactions among various groups of agents in $\mathcal{M}$. Every entry $\mathcal{D}_i = (s_i, \tau_{s_i})$ contains a trajectory $\tau_{s_i}$ for a group of agents $s_i \subseteq N$. Notably, $\tau_{s_i}$ contains the collective trajectory of all agents in the group $s_i$.

**Shapley Values.** We now define the concept of the Shapley Value of an agent (Shapley, 1953), which is commonly used to evaluate the contributions of individual agents to a collective value function in a characteristic function game. Definition 3.2 below is somewhat unconventional but can be easily seen to be equivalent to the standard definition.

**Definition 3.1** (Characteristic function game). A characteristic function game $G$ is given by a pair $(N, v)$, where $N = \{1, \ldots, n\}$ is a finite, non-empty set of agents and $v : 2^N \to \mathbb{R}$ is a characteristic function, which maps each group (sometimes also referred to as coalition) $C \subseteq N$ to a real number $v(C)$; it is assumed that $v(\emptyset) = 0$. The number $v(C)$ is referred to as the value of the group $C$.

Given a characteristic function game $G = (N, v)$, let $\Pi_{N \setminus \{i\}}$ denote the set of all permutations of $N \setminus \{i\}$, i.e., one-to-one mappings from $N \setminus \{i\}$ to itself. For each permutation $\pi \in \Pi_{N \setminus \{i\}}$, we denote by $S_\pi(m)$ the slice of $\pi$ up until and including position $m$; we think of $S_\pi(m)$ as the set of all agents that appear in the first $m$ positions in $\pi$ (note that $S_\pi(0) = \emptyset$). The marginal contribution of an agent $i$ with respect to a permutation $\pi$ and a slice $m$ in a game $G = (N, v)$ is given by

$$\Delta_{m,\pi}^G(i) = v(S_\pi(m) \cup \{i\}) - v(S_\pi(m)).$$

This quantity measures the increase in the value of the group when agent $i$ joins them. We can now define the Shapley Value of an agent $i$: it is simply the agent's average marginal contribution, where the average is taken over all permutations of the set of all other agents $N \setminus \{i\}$ and all slices.

**Definition 3.2** (Shapley Value). Given a characteristic function game $G = (N, v)$ with $|N| = n$, the Shapley Value of an agent $i \in N$ is denoted by $SV_i(G)$ and is given by

$$SV_i(G) = \sfrac{1}{n!} \cdot \sum_{m=0}^{n-1} \sum_{\pi \in \Pi_{N \setminus \{i\}}} \Delta_{m,\pi}^G(i). \tag{1}$$

Def. 3.2 is important in the context of credit assignment, as a possible solution for distributing collective value to individual agents. It also has several consistency properties (Shapley, 1953).

# 4 METHODS

**Problem setting.** Given a dataset $\mathcal{D}$ of trajectories generated by groups of interacting agents and a *Desired Value Function* (DVF), the goal of our work is to learn an imitation policy for a single agent that is aligned with the DVF. We assume that a fraction of the demonstrator agents' behavior is undesirable; specifically, their presence in a group significantly reduces the DVF. Further, we assume that the number of demonstrator agents is much larger than the group size.

**Overview of the methods section.** To evaluate agents' contributions in games that only permit specific group sizes, we first define the concept of EVs (Def. 4.1) for regular characteristic function games (Def. 3.1). We then show that our definition extends naturally to characteristic function games with constraints on permitted group sizes. We finally derive methods to estimate EVs from real-world datasets with limited observations (see Figure 2 for an overview).

## 4.1 EXCHANGE VALUES TO EVALUATE AGENTS' INDIVIDUAL CONTRIBUTIONS

Note that each term of the Shapley Value, denoted $\Delta_{m,\pi}^G(i)$, requires computing the difference in values between two groups of *different* sizes, with and without an agent $i$ (see Def. 3.2). If we wish

to only compare groups with the same size, then a natural alternative is to compute the difference in values when the agent at position $m$ is replaced with agent $i$:

$$\Gamma^G_{m,\pi}(i) = v(S_\pi(m-1) \cup \{i\}) - v(S_\pi(m)). \tag{2}$$

We call this quantity the *exchange contribution* of $i$, given a permutation of agents $\pi$ sliced at position $m$. It represents the added value of agent $i$ in a group. Importantly it does not require values of groups of different sizes.

We now define the EV analogously to the Shapley Value as the average exchange contribution over all permutations of $N \backslash \{i\}$ and all non-empty slices.

**Definition 4.1** (Exchange Value). Given a characteristic function game $G = (N, v)$ with $|N| = n$, the Exchange Value of an agent $i \in N$ is denoted by $EV_i(G)$ and is given by

$$EV_i(G) = ((n-1)! \cdot (n-1))^{-1} \cdot \sum_{m=1}^{n-1} \sum_{\pi \in \Pi_{N \backslash \{i\}}} \Gamma^G_{m,\pi}(i). \tag{3}$$

In words, the EV of an agent can hence be understood as the expected change in value when substituting the agent with another randomly selected agent, or as comparing the value of all groups that include the agent to that of all groups that do not include the agent (see Step 1 in Figure 1).

**Relationship between Shapley Value and Exchange Value.** It can be shown that the Exchange Values of all agents can be derived from their Shapley Values by a simple linear transformation: we subtract a fraction of the value of the grand coalition $N$ (group of all agents) and scale the result by $n/n-1$: $EV_i(G) = \frac{n}{n-1}(SV_i(G) - 1/n \cdot v(N))$. The proof proceeds by computing the coefficient of each term $v(C)$, $C \subseteq N$, in summations (1) and (3) (see Appendix A). Hence, the Shapley Value and the Exchange Value order the agents in the same way. Now, recall that the Shapley Value is characterized by four axioms, namely, dummy, efficiency, symmetry, and linearity (Shapley, 1953). The latter two are also satisfied by the Exchange Value: if $v(C \cup \{i\}) = v(C \cup \{j\})$ for all $C \subseteq N \setminus \{i, j\}$, we have $EV_i(G) = EV_j(G)$ (symmetry), and if we have two games $G_1$ and $G_2$ with characteristic functions $v_1$ and $v_2$ over the same set of agents $N$, then for the combined game $G = (N, v)$ with the characteristic function $v$ given by $v(C) = v_1(C) + v_2(C)$ we have $EV_i(G) = EV_i(G_1) + EV_i(G_2)$ (linearity). The efficiency property of the Shapley Value, i.e., $\sum_{i \in N} SV_i(G) = v(N)$ implies that $\sum_{i \in N} EV_i(G) = 0$. In words, the sum of all agents' EV is zero. The dummy axiom, too, needs to be modified: if an agent $i$ is a dummy, i.e., $v(C \cup \{i\}) = v(C)$ for every $C \subseteq N \setminus \{i\}$ then for the Shapley value we have $SV_i(G) = 0$ and hence $EV_i(G) = -1/n-1 \cdot v(N)$, In each case, the proof follows from the relationship between the Shapley Value and the Exchange Value and the fact that the Shapley Value satisfies these axioms (see Appendix A).

### 4.1.1 COMPUTING EXCHANGE VALUES IF ONLY CERTAIN GROUP SIZES ARE PERMITTED

For a characteristic function game $\mathcal{G} = (N, v)$ the value function $v$ can be evaluated for each possible group $C \subseteq N$. We now consider the case where the value function $v$ is only defined for groups of certain sizes $m \in M$, i.e. $v$ is only defined for a subset of groups of certain sizes.

**Definition 4.2** (Constrained characteristic function game). A constrained characteristic function game $\bar{G}$ is given by a tuple $(N, v, M)$, where $N = \{1, \ldots, n\}$ is a finite, non-empty set of agents, $M \subseteq \{0, \ldots, n-1\}$ is a set of feasible group sizes and $v : \{C \in 2^N : |C| \in M\} \to \mathbb{R}$ is a characteristic function, which maps each group $C \subseteq N$ of size $|C| \in M$ to a real number $v(C)$.

Note that both the Shapley Value and the EV are generally undefined for constrained characteristic function games, as the value function is not defined for groups $C$ of size $|C| \notin M$. The definition of the Shapley Value cannot easily be adapted to constrained characteristic function games, as its computation requires evaluating values of groups of different sizes. In contrast, the definition of the EV can be straightforwardly adapted to constrained characteristic function games by limiting the summation to slices of size $m \in M^+$, where $M^+ = \{m \in M : m > 0\}$. Hence, we define the Constrained EV as the average exchange contribution over all permutations of $N \backslash \{i\}$ and over all slices of size $m \in M^+$.

**Definition 4.3** (Constrained Exchange Value). Given a constrained characteristic function game $\bar{G} = (N, v, M)$ with $|N| = n$, the Constrained Exchange Value of an agent $i \in N$ is denoted by $EV_i(\bar{G})$ and is given by $EV_i(\bar{G}) = ((n-1)! \cdot |M^+|)^{-1} \cdot \sum_{m \in M^+} \sum_{\pi \in \Pi_{N \backslash \{i\}}} \Gamma^{\bar{G}}_{m,\pi}(i)$.

We refer to the Constrained EV and EV interchangeably, as they are applicable to different settings. If some groups are not observed, we can achieve an unbiased estimate of the EV by sampling groups uniformly at random. The expected EV is $EV_i(\bar{G}) = \mathbb{E}_{m \sim U(M^+), \pi \sim U(\Pi_{N \setminus \{i\}})} \left[ \Gamma_{m,\pi}^{\bar{G}}(i) \right]$. This expectation converges to the true EV in the limit of infinite samples.

As outlined in Step 1 in Figure 1, the EV of an agent is a comparison of the value of a group that includes the agent and a group that does not include the agent, considering all permitted group sizes.

## 4.2 ESTIMATING EXCHANGE VALUES FROM LIMITED DATA

The EV assesses the contribution of an individual agent and is applicable under group size limitations in real-world scenarios (see `Group-Limited` in Figure 2). However, exactly calculating EVs is almost always impossible as real-world datasets likely do not contain observations for all (combinatorially many) possible groups (`Low-Data` in Figure 2). We first show a sampling-based estimate (Section 4.2) of EVs, which may have a high variance for EVs of agents that are part of only a few trajectories (outcomes). Next, we introduce a novel method, EV-Clustering (Section 4.2.1), which clusters and can be used to reduce the variance. When datasets are anonymized with one-time-use IDs, each demonstrator is only observed as part of one group (see `Degenerate` in Figure 2), rendering credit assignment degenerate, as explained in Section 4.2.1. We address this by incorporating low-level behavior data from the trajectories $\tau$.

### 4.2.1 EV-CLUSTERING IDENTIFIES SIMILAR AGENTS

In the case of very few agent observations, the above-introduced sampling estimate has a high variance. One way to reduce the variance is by clustering: if we knew that some agents tend to contribute similarly to the DVF, then clustering them and estimating one EV per cluster (instead of one EV per agent) will use more samples and thereby reduce the variance. Note that, as our focus is on accurately estimating EVs, we do not consider clustering agents by behavior here, as two agents may exhibit distinct behaviors while still contributing equally to the DVF.

We propose *EV-Clustering*, which clusters agents such that the variance in EVs across all agents is maximized. In Appendix A we show that *EV-Clustering* is equivalent to clustering agents by their *unobserved* individual contribution, under the approximation that the total value of a group is the sum of the participating agents' individual contributions, an assumption frequently made for theoretical analysis (Lundberg & Lee, 2017; Covert & Lee, 2021), as it represents the simplest nontrivial class of cooperative games. Intuitively, if we choose clusters that maximize the variance in EVs across all agents, all clusters' EVs will be maximally distinct. An example of poor clustering is a random partition, which will have very similar EVs across clusters (having low variance).

Specifically, we assign $n$ agents to $k \leq n$ clusters $K = \{1, \ldots, k-1\}$, with individual cluster assignments $\mathbf{u} = \{u_0, ..., u_{n-1}\}$, where $u_i \in K$. We first combine the observations of all agents within the same cluster by defining a clustered value function $\tilde{v}(C)$ that assigns a value to a group of cluster-centroid agents $C \subseteq K$ by averaging over the combined observations, as $\tilde{v}(C) = 1/\eta \cdot \sum_{m=0}^{n-1} \sum_{\pi \in \Pi_N} v(S_\pi(m)) \cdot \mathbb{1}(\{u_j \mid j \in S_\pi(m)\} = C)$, where $\eta$ is a normalization constant. The EV of an agent $i$ is then given as $EV_i(\tilde{G})$, with $\tilde{G} = (K, \tilde{v})$, thereby assigning equal EVs to all agents within one cluster.

**Definition 4.4** (EV-Clustering). We define the optimal cluster assignments $\mathbf{u}^*$ such that the variance in EVs across all agents is maximized:

$$\mathbf{u}^* \in \arg\max_{\mathbf{u}} \mathrm{Var}([EV_0(\tilde{G}), \ldots, EV_{n-1}(\tilde{G})]). \tag{4}$$

We show in Appendix B.1 that this objective is equivalent to clustering agents by their unobserved individual contributions, under the approximation of an additive value function.

### 4.2.2 DEGENERACY OF THE CREDIT ASSIGNMENT PROBLEM FOR FULLY-ANONYMIZED DATA

If two agents are observed only once in the dataset and as part of the same group, equal credit must be assigned to both due to the inability to separate their contributions. Analogously, when all agents are only observed once, credit can only be assigned to groups, resulting in the degenerate scenario

Table 1: Resulting performance with respect to the DVF for different imitation learning methods in different Starcraft scenarios.

| Method | 2s3z | 3s_vs_5z | 6h_vs_8z |
|---|---|---|---|
| BC | $12.14 \pm 1.8$ | $13.10 \pm 2.0$ | $8.56 \pm 0.6$ |
| Group-BC | $15.41 \pm 2.4$ | $16.63 \pm 1.9$ | $9.10 \pm 0.9$ |
| EV2BC (Ours) | $\mathbf{17.38} \pm 1.6$ | $\mathbf{20.31} \pm 2.4$ | $\mathbf{10.0} \pm 0.91$ |

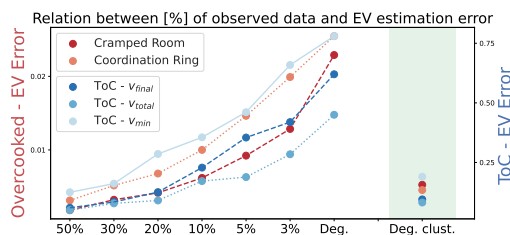

Figure 3: Mean error in estimating EVs with decreasing number of observations. 'Deg.' refers to the fully anonymized degenerate case. Error decreases significantly if agents are clustered (green-shaded area).

that all agents in a group are assigned the same credit (e.g. are assigned equal EV). We solve this by combining low-level behavior information from trajectories $\tau$ with EV-Clustering (see Sec. 5.1).

### 4.3 Exchange Value based Behavior Cloning (EV2BC)

Having defined the EV of an individual agent and different methods to estimate it, we now define a variation of Behavior Cloning (Pomerleau, 1991), which takes into account each agent's contribution to the desirability value function (DVF). We refer to this method as *EV2BC*. Essentially, EV2BC imitates only actions of selected agents that have an EV larger than a tunable threshold parameter.

**Definition 4.5** (EV based Behavior Cloning (EV2BC)). For a set of demonstrator agents $N$, a dataset $\mathcal{D}$, and a DVF, we define the imitation learning loss for EV2BC as

$$L_{EV2BC}(\theta) = -\sum_{n \in \mathcal{N}} \sum_{(s_i, a_i^n) \in \mathcal{D}} \log(\pi^\theta(a_i^n | s_i)) \cdot \mathbb{1}(EV_n^{DVF} > c) \tag{5}$$

where $EV_n^{DVF}$ denotes the EV of agent $n$ and where $c$ is a tunable threshold parameter that trades off between including data of agents with higher contributions to the DVF and reducing the total amount of training data used.

## 5 Experiments

The environments that we consider only permit certain group sizes, hence we use constrained EVs (see Def. 4.3). We run all experiments for five random seeds and report mean and standard deviation where applicable. For more details on the implementation, please refer to the Appendix. In the following experiments, we first evaluate EVs as a measure of an agent's contribution to a given DVF. We then assess the average estimation error for EVs as the number of observations in the dataset $D$ decreases and how applying clustering decreases this error. We lastly evaluate the performance of Exchange Value based Behaviour Cloning (EV2BC, see Definition 4.5) for simulated and human datasets and compare to relevant baselines, such as standard Behavior Cloning (Pomerleau, 1991) and Offline Reinforcement Learning (Pan et al., 2022).

In **Tragedy of the Commons** (Hardin, 1968) (ToC) multiple individuals deplete a shared resource. It is a social dilemma scenario often studied to model the overexploitation of common resources (Dietz et al., 2003; Ostrom, 2009). We model ToC as a multi-agent environment and consider three DVFs representing different measures of social welfare: the final pool size $v_{final}$, the total resources consumed $v_{total}$, and the minimum consumption among agents $v_{min}$.

**Overcooked** (Carroll et al., 2019) is a two-player game simulating a cooperative cooking task requiring coordination and is a common testbed in multi-agent research. Within Overcooked, we consider the configurations Cramped Room and Coordination Ring (displayed in Figure 4). For each environment configuration, we generate two datasets by simulating agent behaviors using a near-optimal planning algorithm, where we use a parameter $\lambda$ to determine an agent's behavior. For $\lambda = 1$ agents act (near)-optimal, for $\lambda = -1$ agents act adversarially. We refer to $\lambda$ as the agent's trait, as it acts as a proxy for the agent's individual contribution to the collective value function. Each demonstration dataset $D$ is generated by $n = 100$ agents, and trajectories $\tau$ are of length 400. The adversarial dataset $D^{adv}$ is comprised of 25% adversarial agents with $\lambda = -1$ and 75% (near)-optimal agents

with $\lambda = 1$, while for the dataset $D^\lambda$ agents were uniformly sampled between $\lambda = -1$ and $\lambda = 1$. The $D^{\text{human}}$ dataset was collected from humans playing the game (see Carroll et al. (2019)); it is fully anonymized with one-time-use agent identifiers, hence is a degenerate dataset (see Figure 2 bottom row). We consider the standard value function given for Overcooked as the DVF, i.e. the number of soups prepared by both agents over the course of a trajectory.

The **StarCraft Multi-Agent Challenge** (Samvelyan et al., 2019) is a cooperative multi-agent environment that is partially observable, involves long-term planning, requires strong coordination, and is heterogeneous. We consider the settings `2s3z`, `3s_vs_5z` and `6h_vs_8z`, which involve teams of 3-6 agents. For each, we generate a pool of 200 agents with varying capabilities by extracting policies at different epochs, and from training with different seeds. We generate a dataset that contains simulated trajectories of 100 randomly sampled groups (out of $10^9$ possible groups) and use the environment's ground truth reward function to assign DVF scores according to the collective performance of agents.

**Exchange Values assess an agent's individual contribution to a collective value function.** To analyze EVs as a measure for an agent's individual contribution to a DVF, we consider full datasets that contain demonstrations of all possible groups, which allows us to estimate EVs accurately. In ToC, we find that the ordering of agents broadly reflects our intuition: Taking more resources negatively impacts the EVs, and agents consuming the average of others have less extreme EVs. The color-coded ordering of agents under different DVFs in shown in Figure 7 in App. C. In Overcooked, we consider the two simulated datasets ($D^{adv}$ and $D^\lambda$) but not the human dataset, as the individual contribution is unknown for human participants. We find that EVs of individual agents are strongly correlated with their trait parameter $\lambda$, which is a proxy for the agent's individual contribution, and provide a plot that shows the relationship between $\lambda$ and EV in Figue 5 in App. B.

### 5.1 Estimating EVs from incomplete data

**Estimation error for different dataset sizes.** We now turn to realistic settings with missing data, where EVs must be estimated (Sec. 4.2). For both ToC and Overcooked, we compute the mean estimation error in EVs if only a fraction of the possible groups is contained in the dataset. As expected, we observe in Fig. 3 that the mean estimation error increases as the fraction of observed groups decreases, with the largest estimation error for fully anonymized datasets (see Fig. 3 – *Deg.*).

**Estimating EVs from degenerate datasets with EV-Clustering.** To estimate EVs from degenerate datasets, we first obtain behavior embeddings from the low-level behavior information given in the trajectories $\tau$ in $D$. Specifically, in Overcooked and ToC, we concatenate action frequencies in frequently observed states. In Starcraft, we use TF-IDF (Spärck Jones, 1972) to obtain behavior embeddings. We then compute a large number of possible cluster assignments for the behavior embeddings using different methods and hyperparameters. In accordance with the objective of EV-Clustering, we choose the cluster assignment with the highest variance in EVs. We observe in Figure 3 that clustering significantly decreases the estimation error (see Deg. clustered).

### 5.2 Imitating desired behavior by utilizing EVs

We now evaluate EV2BC in all domains. In accordance with the quantity of available data, we set the threshold parameter such that only agents with EVs above the 90th, 67th, and 50th percentile are imitated in ToC, Starcraft, and Overcooked, respectively. We replicate the single-agent EV2BC policy for all agents in the environment and evaluate the achieved collective DVF score. As baselines, we consider (1) *BC*, where Behavior Cloning (Pomerleau, 1991) is done with the full dataset without correcting for EVs, (2) offline multi-agent reinforcement *OMAR* (Pan et al., 2022) with the reward at the last timestep set to the DVF's score for a given trajectory (no per-step reward is given by the DVF) and (3) *Group BC*, for which only collective trajectories with a DVF score larger than the relevant percentile are included. While EV2BC is based on individual agents' contributions, this last baseline selectively imitates data based on group outcomes. For instance, if a collective trajectory includes two aligned agents and one unaligned agent, the latter baseline is likely to imitate all three agents. In contrast, our approach would only imitate the two aligned agents.

Table 2: Resulting performance with respect to the DVF for different imitation learning methods in the Overcooked environments Cramped Room (top) and Coordination Ring (bottom). In Tragedy of Commons: 12 agents experiment at the top, 120 agents experiment at the bottom.

| Imitation method | Overcooked | | | Tragedy of Commons | | |
|---|---|---|---|---|---|---|
| | $\mathcal{D}^{\lambda}$ | $\mathcal{D}^{adv}$ | $\mathcal{D}^{\text{human}}$ | $v_{final}$ | $v_{total}$ | $v_{min}$ |
| BC (Pomerleau, 1991) | $10.8 \pm 2.14$ | $40.8 \pm 12.7$ | $153.34 \pm 11.5$ | $2693.6 \pm 139.1$ | $50.6 \pm$ | $2.4 \pm 0.45$ |
| Group-BC | $54.2 \pm 5.45$ | $64.8 \pm 7.62$ | $163.34 \pm 6.08$ | $5324.2 \pm 210.8$ | $100.01 \pm 20.08$ | $4.60 \pm 1.01$ |
| OMAR (Pan et al., 2022) | $6.4 \pm 3.2$ | $25.6 \pm 8.9$ | $12.5 \pm 4.5$ | - | - | - |
| EV2BC (ours) | $\mathbf{91.6} \pm 12.07$ | $\mathbf{104.2} \pm 10.28$ | $\mathbf{170.89} \pm 6.8$ | $\mathbf{10576.8} \pm 307.4$ | $\mathbf{342.8} \pm 49.36$ | $\mathbf{44.2} \pm 6.4$ |
| BC (Pomerleau, 1991) | $15.43 \pm 4.48$ | $10.4 \pm 6.8$ | $104.89 \pm 12.44$ | $2028.8 \pm 60.9$ | $38.9 \pm 10.4$ | $1.8 \pm 0.4$ |
| Group-BC | $24 \pm 4.69$ | $\mathbf{14.6} \pm 2.48$ | $102.2 \pm 6.19$ | $3400.5 \pm 100.9$ | $77.1 \pm 14.1$ | $3.51 \pm 1.6$ |
| OMAR (Pan et al., 2022) | $12.43 \pm 3.35$ | $9.5 \pm 3.5$ | $12.4 \pm 6.0$ | - | - | - |
| EV2BC (ours) | $\mathbf{30.2} \pm 6.91$ | $12.4 \pm 2.65$ | $\mathbf{114.89} \pm 5.08$ | $\mathbf{8123.4} \pm 600.8$ | $\mathbf{270.0} \pm 50.0$ | $\mathbf{33.1} \pm 7.1$ |

**ToC results.** We imitate datasets of 12 agents and 120 agents, with group sizes of 3 and 10, respectively, evaluating performance for each of the three DVFs defined for the ToC environment. We do not consider the OMAR baseline as policies are not learned but rule-based. Table 1 demonstrates that EV2BC outperforms the baselines by a large margin.

**Overcooked results.** We now consider all datasets $D^{adv}$, $D^{\lambda}$ and $D^{\text{human}}$ in both Overcooked environments. We evaluate the performance achieved by agents with respect to the DVF (the environment value function of maximizing the number of soups) when trained with different imitation learning approaches on the different datasets. EVs are computed as detailed in Section 5.1. Table 1 shows that EV2BC clearly outperforms the baseline approaches. We further note that EV2BC significantly outperforms baseline approaches on the datasets of human-generated behavior, for which EVs were estimated from a fully-anonymized real-world dataset. This demonstrates that BC on datasets containing unaligned behavior carries the risk of learning wrong behavior, but it can be alleviated by weighting the samples using estimated EVs.

**Starcraft Results.** We observe in Table 1 that EV2BC outperforms the baselines by a substantial margin, underlining the applicability of our method to larger and more complex settings. We omitted the OMAR baseline, which is implemented as offline MARL with the DVF as the final-timestep reward, as it performed significantly worse than BC.

## 6 CONCLUSION

Our work presents a method for training AI agents from diverse datasets of human interactions while ensuring that the resulting policy is aligned with a given desirability value function. However, it must be noted that quantifying this value function is an active research area. Shapley Values and Exchange Values estimate the alignment of an individual with a group value function (which must be prescribed separately) and, as such, can be misused if they are included in a larger system that is used to judge those individuals in any way. Discrimination of individuals based on protected attributes is generally unlawful, and care must be taken to avoid any discrimination by automated means. We demonstrated a novel positive use of these methods by using them to train aligned (beneficial) agents, that do not imitate negative behaviors in a dataset. We expect that the benefits of addressing the problem of unsafe behavior by AI agents outweigh the downsides of misuse of Shapley Values and Exchange Values, which are covered by existing laws.

Future work may address the assumption that individual agents behave similarly across multiple trajectories and develop methods for a more fine-grained assessment of desired behavior. Additionally, exploring how our framework can more effectively utilize data on undesired behavior is an interesting avenue for further investigation, e.g., developing policies that are constrained to not taking undesirable actions. Lastly, future work may investigate applications to real-world domains, such as multi-agent autonomy scenarios.

**Reproducibility.** To help reproduce our work, we publish code on the project website at `https://tinyurl.com/select-to-perfect`. We provide detailed overviews for all steps of the experimental evaluation in the Appendix, where we also link to the publicly available code repositories that our work used. We further provide information about computational complexity at the end of the Appendix.

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

## A APPENDIX TO METHODS

### A.1 AXIOMATIC PROPERTIES OF THE EXCHANGE VALUE AND ITS RELATIONSHIP WITH THE SHAPLEY VALUE

Fix a characteristic function game $G$ with a set of players $N$. It is well-known that the Shapley Value satisfies the following axioms (Shapley, 1953):

(1) Dummy: if an agent $i$ satisfies $v(C \cup \{i\}) = v(C)$ for all $C \subseteq N \setminus \{i\}$ then $SV_i(G) = 0$;

(2) Efficiency: the sum of all agents' Shapley Values equals to the value of the grand coalition, i.e., $\sum_{i \in N} SV_i(G) = v(N)$;

(3) Symmetry: for every pair of distinct agents $i, j \in N$ with $v(C \cup \{i\}) = v(C \cup \{j\})$ for all $C \subseteq N \setminus \{i, j\}$ we have $SV_i(G) = SV_j(G)$;

(4) Linearity: for any pair of games $G_1 = (N, v_1)$ and $G_2 = (N, v_2)$ with the same set of agents $N$, the game $G = (N, v)$ whose characteristic funciton $v$ is given by $v(C) = v_1(C) + v_2(C)$ for all $C \subseteq N$ satisfies $SV_i(G) = SV_i(G_1) + SV_i(G_2)$ for all $i \in N$.

Indeed, the Shapley Value is the only value for characteristic function games that satisfies these axioms (Shapley, 1953). It is then natural to ask which of these axioms (or their variants) are satisfied by the Exchange Value.

To answer this question, we first establish a relationship between the Shapley Value and the Exchange Value.

**Proposition A.1.** *For any characteristic function game $G = (N, v)$ and every agent $i \in N$ we have*

$$EV_i(G) = \frac{n}{n-1} \left( SV_i(G) - \frac{1}{n} \cdot v(N) \right). \tag{6}$$

*Proof.* Fix an agent $i$ and consider an arbitrary non-empty coalition $C \subsetneq N \setminus \{i\}$.

In the expression for the Shapley Value of $i$ the coefficient of $v(C)$ is

$$-\frac{1}{n!}(|C|)!(n-1-|C|)! :$$

we subtract the fraction of permutations of $N$ where the agents in $C$ appear in the first $|C|$ positions, followed by $i$. By the same argument, the coefficient of $v(C \cup \{i\})$ is

$$\frac{1}{n!}(|C|)!(n-1-|C|)!.$$

Similarly, in the expression for the Exchange Value of $i$ the coefficient of $v(C)$ is

$$-\frac{1}{(n-1)!(n-1)}(|C|)!(n-1-|C|)! :$$

each permutation of $N \setminus \{i\}$ where agents in $C$ appear in the first $|C|$ positions contributes with coefficient $-\frac{1}{(n-1)!(n-1)}$. By the same argument, the coefficient of $v(C \cup \{i\})$ is

$$\frac{1}{(n-1)!(n-1)}(|C|)!(n-1-|C|)!$$

Now, if $C = N \setminus \{i\}$, in the expression for $SV_i(G)$ the coefficient of $v(C)$ is $-\frac{1}{n}$ and the coefficient of $v(C \cup \{i\}) = v(N)$ is $\frac{1}{n}$. In contrast, in the expression for $EV_i(G)$ the coefficient of $v(C)$ is $-\frac{1}{n-1}$: for each of the $(n-1)!$ permutations of $N \setminus \{i\}$ we subtract $v(C)$ with coefficient $\frac{1}{(n-1)!(n-1)}$ when we replace the last agent in that permutation by $i$. On the other hand, $v(N)$ never appears.

It follows that, for every coalition $C \subsetneq N$, if the value $v(C)$ appears in the expression for $SH_i(G)$ with weight $\omega$ then it appears in the expression for $EV_i(G)$ with weight $\frac{n}{n-1} \cdot \omega$. Hence

$$EV_i(G) = \frac{n}{n-1} \left( SH_i(G) - \frac{1}{n} \cdot v(N) \right)$$

$\square$

**Example A.2.** *Consider a characteristic function game $G = (N, v)$, where $N = \{1, 2\}$ and $v$ is given by $v(\{1\}) = 2$, $v(\{2\}) = 4$, $v(\{1, 2\}) = 10$. We have*

$$SH_1(G) = (2 + (10 - 4))/2 = 4, \qquad SH_2(G) = (4 + (10 - 2))/2 = 6$$

*and*

$$EV_1(G) = 2 - 4 = -2, \qquad EV_2(G) = 4 - 2 = 2.$$

*Note that $EV_1(G) = 2(SH_1(G) - \frac{1}{2}v(N))$, $EV_2(G) = 2(SH_2(G) - \frac{1}{2}v(N))$.*

We can use Proposition A.1 to show that the Exchange Value satisfies two of the axioms satisfied by the Shapley Value, namely, linearity and symmetry.

**Proposition A.3.** *The Exchange Value satisfies symmetry and linearity axioms.*

*Proof.* For the symmetry axiom, fix a characteristic function game $G = (N, v)$ and consider two agents $i, j \in N$ with $v(C \cup \{i\}) = v(C \cup \{j\})$ for all $C \subseteq N \setminus \{i, j\}$. We have

$$EV_i(G) = \frac{n}{n-1}\left(SV_i(G) - \frac{1}{n} \cdot v(N)\right) = \frac{n}{n-1}\left(SV_j(G) - \frac{1}{n} \cdot v(N)\right) = EV_j(G),$$

where the first and the third equality follow by Proposition A.1, and the second equality follows because the Shapley Value satisfies symmetry.

For the linearity axiom, consider a pair of games $G_1 = (N, v_1)$ and $G_2 = (N, v_2)$ with the same set of agents $N$ and the game $G = (N, v)$ whose characteristic funciton $v$ is given by $v(C) = v_1(C) + v_2(C)$ for all $C \subseteq N$. Fix an agent $i \in N$. We have

$$EV_i(G) = \frac{n}{n-1}\left(SV_i(G) - \frac{1}{n} \cdot (v_1(N) + v_2(N))\right)$$

$$= \frac{n}{n-1}\left(SV_i(G_1) - \frac{1}{n} \cdot v_1(N)\right) + \frac{n}{n-1}\left(SV_i(G_2) - \frac{1}{n} \cdot v_2(N)\right)$$

$$= EV_i(G_1) + EV_i(G_2).$$

Again, the first and the third equality follow by Proposition A.1, and the second equality follows because the Shapley Value satisfies linearity. $\square$

While the Exchange Value does not satisfy the dummy axiom or the efficiency axiom, it satisfies appropriately modified versions of these axioms.

**Proposition A.4.** *For every characteristic function game $G$ it holds that $\sum_{i \in N} EV_i(G) = 0$. Moreover, if $i$ is a dummy agent, i.e., $v(C \cup \{i\}) = V(C)$ for all $C \subseteq N \setminus \{i\}$ then $EV_i(G) = -\frac{v(N)}{n-1}$.*

*Proof.* We have

$$\sum_{i \in N} EV_i(G) = \sum_{i \in N} \frac{n}{n-1}\left(SV_i(G) - \frac{1}{n} \cdot v(N)\right) = \sum_{i \in N} \frac{n}{n-1} SV_i(G) - \frac{n}{n-1} \cdot v(N)$$

$$= \frac{n}{n-1} \cdot v(N) - \frac{n}{n-1} \cdot v(N) = 0,$$

where we use Proposition A.1 and the fact that the Shapley Value satisfies the efficiency axiom.

Now, fix a dummy agent $i$. We have

$$EV_i(G) = \frac{n}{n-1}\left(SV_i(G) - \frac{1}{n} \cdot v(N)\right) = -\frac{1}{n-1} \cdot v(N);$$

again, we use Proposition A.1 and the fact that the Shapley Value satisfies the dummy axiom. $\square$

## A.2 DERIVATION OF CLUSTERING OBJECTIVE STATED IN EQ. 4

**Inessential games and EVs.** The assumption of an inessential game is commonly made to compute Shapley Values more efficiently[2]. In an inessential game, the value of a group is given by the sum of the individual contributions of its members, denoted as $v(C) = \sum_{i \in C} v_i$, where $v_i$ is an individual agent's unobserved contribution $v_i$. The EV (see Definition 4.1) of an individual agent $i$ in an inessential game is given as

$$EV_i(G) = v_i - \frac{1}{|N|-1} \cdot \sum_{j \in N \setminus \{i\}} v_j = (1 + \frac{1}{|N|-1}) \cdot v_i - \frac{1}{|N|-1} \cdot \sum_{j \in N} v_j,$$

This expression represents the difference between the individual contribution of agent $i$, $v_i$, and the average individual contribution of all other agents. The second term is independent of $i$ and remains constant across all agents.

---

[2]see, e.g., Covert, I. and Lee, S.I., 2020. Improving kernelshap: Practical shapley value estimation via linear regression. arXiv preprint arXiv:2012.01536.

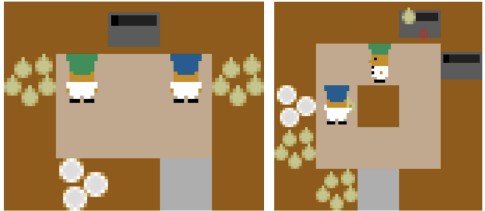

Figure 4: In the Overcooked environments Cramped Room (left) and Coordination Ring (right), agents must cooperate to cook and deliver as many soups as possible within a given time.

**Derivation of equivalent clustering objective.** We now consider the optimization problem defined by Equation 4, which defines optimal cluster assignments $\mathbf{u}^*$ such that the variance in EVs is maximised

$$\mathbf{u}^* \in \arg\max_{\mathbf{u}} \text{Var}([\tilde{EV}_0(\tilde{G}), \ldots, \tilde{EV}_{n-1}(\tilde{G})]).$$

Further, the clustered value function is defined as

$$\tilde{v}(C) = \frac{1}{\eta} \cdot \sum_{m=0}^{n-1} \sum_{\pi \in \Pi_N} v(S_\pi(m)) \cdot \mathbb{1}(\{u_j \mid j \in S_\pi(m)\} = C),$$

where the normalisation constant is defined as $\eta = \sum_{m=0}^{n-1} \sum_{\pi \in \Pi_N} \mathbb{1}(\{u_j \mid j \in S_\pi(m)\} = C)$. We denote by $k_i$ the individual contribution of the agent that represents the agents in cluster $i$. The value $k_i$ is defined as the average individual contribution of all agents assigned to the cluster, i.e. $k_i = \frac{1}{\epsilon} \cdot \sum_{j \in N} v_j \cdot \mathbb{1}(\mathbf{u}(i) = \mathbf{u}(j))$. Here, the normalization constant is given as $\epsilon = \sum_{j \in N} \mathbb{1}(\mathbf{u}(i) = \mathbf{u}(j))$.

Using the concept of the clustered value function $\tilde{v}$, we can express the EV for all agents assigned cluster $i$ as

$$EV_i(\tilde{G}) = (1 + \frac{1}{|K|-1}) \cdot k_i - \frac{1}{|K|-1} \cdot \sum_{j \in K} k_j.$$

The second term, which is cluster-independent, can be omitted when computing the variance $\text{Var}([EV_0(\tilde{G}), \ldots, EV_{n-1}(\tilde{G})])$, as the variance is agnostic to a shift in the data distribution. We will omit the scaling factor $(1 + \frac{1}{|K|-1})$ from here onwards.

Let $n_j$ denote the number of agents assigned to cluster $j \in K$, with $\sum_{i=0}^{K-1} n_i = N$. By simplifying Equation 4, we obtain:

$$\text{Var}([EV_0(\tilde{G}), \ldots, EV_{n-1}(\tilde{G})]) = \sum_{i=0}^{K-1} n_i \cdot \left(k_i - \frac{\sum_{j=0}^{K-1} n_j \cdot k_j}{N}\right)^2.$$

This allows us to express the objective stated in Equation 4 as

$$\mathbf{u}^* \in \arg\max_{\mathbf{u}} \text{Var}([k_0, \ldots, k_{n-1}]).$$

The objective stated in Equation 4 is therefore equivalent to assigning agents to clusters such that the variance in cluster centroids (centroids computed as the mean of the unobserved individual contributions $v_i$ of all agents assigned to a given cluster) is maximized.

Table 3: Dataset statistics in Overcooked.

| Imitation method | Cramped Room $D^\lambda$ | Coordination Ring $D^\lambda$ | Cramped Room $D^{adv}$ | Coordination Ring $D^{adv}$ |
|---|---|---|---|---|
| Minimum | 0 | 0 | 0 | 0 |
| Mean | $20.6 \pm 33.58$ | $12 \pm 19.39$ | $16.91 \pm 40.64$ | $3 \pm 11.15$ |
| Maximum | 150 | 80 | 160 | 80 |

# B OVERCOOKED EXPERIMENTS

We generate the simulated datasets using the planning algorithm given in Carroll et al. (2019)[3]. To be able to simulate agents with different behaviors (from adversarial to optimal), we first introduce a latent trait parameter,

---

[3] https://github.com/HumanCompatibleAI/overcooked_ai

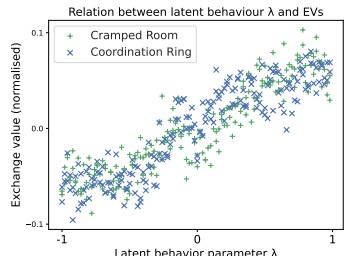

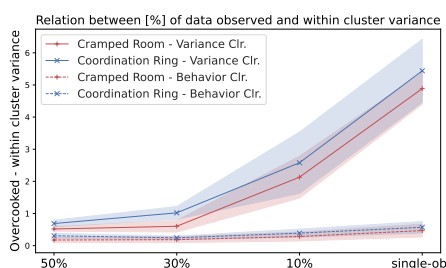

Figure 5: Relation between an agent's trait $\lambda$ and its EV in Overcooked.

Figure 6: Within-cluster variance in relation to fraction of observations for simulated data in Cramped Room and Coordination Ring (Overcooked). Two clustering methods shown (Behavior clustering and Variance Clustering). In the case of random cluster assignments, the within-cluster variance is $5.11 \pm 0.11$, while under optimal cluster assignments, the variance is $0.156$. See section B.1 for discussion.

$\lambda$, which determines the level of adversarial or optimal actions for a given agent. A value of $\lambda = 1$ represents a policy that always chooses the best action with certainty. As $\lambda$ decreases, agents are more likely to select non-optimal actions. For $\lambda < 0$, we invert the cost function to create agents with adversarial behavior. Notably, we assign a high cost (or low cost when inverted) to occupying the cell next to the counter in the Overcooked environment. Occupying the cell next to the counter enables adversarial agents to block other agents in the execution of their tasks.

For human gameplay datasets, we utilized the raw versions of the Overcooked datasets.[4] These datasets were used as-is, without manual pre-filtering.

**EVs.** To estimate agents' EVs according to Section 4.2, we used either the full set of all possible groups or a fraction of it (see Figure 3 for the relationship between dataset size and EV estimation error). For each observed grouping, we conducted 10 rollouts in the environment and calculated the average score across these rollouts to account for stochasticity in the environment.

**Imitation learning.** For EV2BC, BC, and group-BC, we used the implementation of Behavior Cloning in Overcooked as given by the authors of (Carroll et al., 2019)[5]. We implement the offline multi-agent reinforcement learning method OMAR (Pan et al., 2022) using the author's implementation.[6] For the OMAR baseline, we set the reward at the last timestep to the DVF's score for a given trajectory, as our work assumes that no per-step reward signal is given, in contrast to the standard offline-RL framework. We conducted a hyperparameter sweep for the following parameters: learning rate with options $\{0.01, 0.001, 0.0001\}$, Omar-coe with options $\{0.1, 1, 10\}$, Omar-iters with options $\{1, 3, 10\}$, and Omar-sigma with options $\{1, 2, 3\}$. The best-performing parameters were selected based on the evaluation results.

### B.1 CLUSTERING OF AGENTS IN OVERCOOKED

**Behavior clustering.** The behavior clustering process in the Overcooked environment involves the following steps. Initially, we identify the 200 states that are most frequently visited by all agents in the given set of observations. As the action space in Overcooked is relatively small ($\leq 7$ actions), we calculate the empirical action distribution for each state for every agent. These 200 action distributions are then concatenated to form a behavior embedding for each agent. To reduce the dimensionality of the embedding, we apply Principal Component Analysis (PCA), transforming the initial embedding space into three dimensions. Subsequently, we employ the k-means clustering algorithm to assign agents to behavior clusters. The number of clusters (3 for Overcooked) is determined using the ELBOW method (Thorndike, 1953), while linear kernels are utilized for both PCA and k-means. It is noteworthy that the results are found to be relatively insensitive to the parameters used in the dimensionality reduction and clustering steps, thus standard implementations are employed for both

---

[4]https://github.com/HumanCompatibleAI/human_aware_rl/tree/master/human_aware_rl/data/human/anonymized

[5]https://github.com/HumanCompatibleAI/overcooked_ai/tree/master/src/human_aware_rl/imitation

[6]https://github.com/ling-pan/OMAR

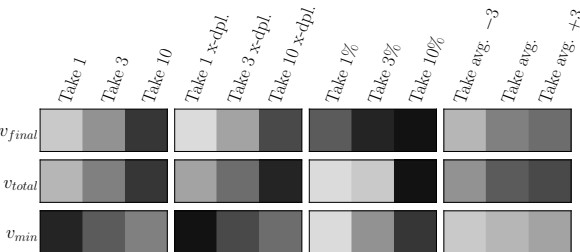

Figure 7: Colour-coded ordering of EVs for agents with varying behaviors in Tragedy of the Commons. The brighter, the higher an agent's contribution to a given value function.

methods (Pedregosa et al., 2011). Importantly, this clustering procedure focuses exclusively on the observed behavior of agents, specifically the actions taken in specific states, and is independent of the scores assigned to trajectories by the DVF.

**EV-Clustering.** In contrast to behavior clustering, EV-Clustering (see Section 4.2.1) focuses solely on the scores assigned to trajectories by the DVF and disregards agent behavior. The objective of variance clustering is to maximize the variance in assigned EVs, as stated in Equation 4. To optimize this objective, we utilize the SLSQP non-linear constrained optimization introduced by Kraft (1988).

We use soft cluster assignments and enforce constraints to ensure that the total probability is equal to one for each agent. The solver is initialized with a uniform distribution and runs until convergence or for a maximum of 100 steps. Given that the optimization problem may have local minima, we perform 500 random initializations and optimizations, selecting the solution with the lowest loss (i.e. the highest variance in assigned EVs).

**Combining Behavior Clustering and EV Clustering.** As described in Sections 4.2.2 and 5.1, behavior clustering (which utilizes behavior information but disregards DVF scores) and variance clustering (which utilizes DVF scores but disregards behavior information) are combined to estimate EVs for degenerate datasets. We initialize the SLSQP solver with the cluster assignments obtained from behavior clustering and introduce a small loss term in the objective function of Equation 4. This additional loss term, weighted by 0.1 (selected in a small sensitivity analysis), penalizes deviations from the behavior clusters. Similar to before, we perform 500 iterations while introducing a small amount of noise to the initial cluster assignments at each step. The solution with the highest variance in assigned EVs is then selected.

**Ablation study.** We present an ablation study to examine the impact of different components in the clustering approach discussed in Section 5.1. We proposed two sequential clustering methods: behavior clustering and variance clustering. This ablation study investigates the performance of both clustering steps when performed independently, also under the consideration of the fraction of the data that is observed. We assess performance as the within-cluster variance in the unobserved agent-specific latent trait variable $\lambda$, where lower within-cluster variance indicates higher performance. It is important to note that $\lambda$ is solely used for evaluating the clustering steps and not utilized during the clustering process. The results of the ablation study are depicted in Figure 6.

We first discuss EV-Clustering. EV-Clustering as introduced in Seciton 4 generally leads to a significant decrease in within-cluster variance in the unobserved variable $\lambda$. More specifically, the proposed variance clustering approach (when 50% of data is observed), results in a $\sim 89\%$ reduction of the within-cluster variance in $\lambda$, which validates the approach of clustering agents by their unobserved individual contributions by maximizing the variance in estimated EVs. However, we observe in Figure 6 that as the fraction of observed data decreases, the within-cluster variance increases, indicating a decrease in the quality of clustering. The highest within-cluster variance is observed when using only a single observation ('single-obs'), which corresponds to a fully-anonymized dataset. This finding is consistent with the fact that a fully-anonymized dataset presents a degenerate credit assignment problem, as discussed in Section 4.2.2.

We now discuss behavior clustering. Figure 6 shows that behavior clustering generally results in a very low within-cluster variance. However, it is important to note that these results may not directly translate to real-world data, as the ablation study uses simulated trajectories. Note that such an ablation study cannot be conducted for the given real-world human datasets, as these are fully anonymized.

## C    TRAGEDY OF THE COMMONS EXPERIMENTS

**Clustering.**    We model ToC as a multi-agent environment where agents consume from a common pool of resources $x_t$, which grows at a fixed rate $g = 25\%$ at each time step $t$: $x_{t+1} = \max\left((1+g) \cdot x_t - \sum_i c_{ti}, \ 0\right)$, with $c_{ti}$ as the consumption of the $i$th agent at time $t$ and $x_0 = 200$ as the starting pool. Hence, if all resources are consumed, none can regrow and no agents can consume more resources. The Tragedy of the Commons (ToC) environment features 4 different behavior patterns: *Take-X* consumes X units at every timestep, *Take-X-x-dpl* consumes X units if this does not deplete the pool of resources, *Take X%* consumes X% of the available resources, and *TakeAvg* consumes the average of the resources consumed by the other agents at the previous timestep (0 in the first timestep). For the small-scale experiment of 12 agents, we consider three agents for each pattern, with X values selected from the set $1, 3, 10$. For the large-scale experiment of 120 agents, we simply replicate each agent configuration 10 times. We simulate both experiments for groups of size 3 and 10, respectively. We generate a simulated dataset using agents with four different behavior patterns. We first collect a dataset of observations for a small-scale experiment of 12 agents and simulate ToC for groups of three agents for 50 time steps (we later consider a group of 120 agents).

Due to the continuous nature of the state and action spaces in ToC, we first discretize both and then apply the same clustering methods used in the Overcooked scenario. We proceed by computing EVs for all agents as done in Overcooked (see Figure 3 for results). We implement imitation policies by replicating the averaged action distributions in the discretized states.

### C.1    COMPUTATIONAL DEMAND.

We used an Intel(R) Xeon(R) Silver 4116 CPU and an NVIDIA GeForce GTX 1080 Ti (only for training BC, EV2BC, group-BC, and OMAR policies). In Overcooked, generating a dataset took a maximum of three hours, and estimating EVs from a given dataset takes a few seconds. Behavior clustering consumes a couple of minutes, while Variance clustering takes up to two hours per configuration (note that it is run 500 times). Training of the BC, group-BC, and EV2BC policies took no more than 30 minutes (using a GPU), while the OMAR baseline was trained for up to 2 hours. In Tragedy of Commons, each rollout only consumes a couple of seconds. Clustering times were comparable to those in Overcooked. Computing imitation policies is similarly only a matter of a few minutes.

## D    STARCRAFT EXPERIMENTS

**Dataset generation.**    For each environment configuration, we first train agents with the ground truth reward function for different seeds. We then extract agents at three different timesteps (0%, 50%, and 100%) of the training. We randomly sample groups of the extracted agents to generate a large set of trajectory rollouts and record the average score achieved by a group, which serves as the observed DVF score. We anonymize the dataset by assigning one-time-use IDs to all agents.

**Clustering of agents and EV computation.**    We first extract high-dimensional features from agent trajectories using Term Frequency-Inverse Document Frequency (TF-IDF (Spärck Jones, 1972)) and compute agent clusters. We compute a large number of possible cluster assignments using different hyperparameters for TF-IDF and different hyperparameters for spectral clustering. In accordance with the objective of EV clustering, we then choose the cluster assignment that results in the largest variance in EVs across all agents.

For each individual agent, we extract the action sequence (up to a cutoff of either 1000 or 50000 environment steps). We then convert the action sequence into a string to apply TD-IDF. This can be thought of as representing each action as an individual word. This results in a corpus that contains one document per agent.

For the transformation of the corpus into a feature space, we apply the TF-IDF vectorization, parameterized by:

- Min_df: Minimum document frequency set at $0.05$, ensuring inclusion of terms present in at least $5\%$ of the documents.
- Max_df: Maximum document frequencies tested were $0.3$ and $0.9$, to filter out terms excessively common across documents.
- Max_features: The upper limit on the number of features considered were set to $100,000$ or $1,000,000$.
- Ngram_range: We explored n-gram ranges of $(1,3)$, $(1,5)$, and $(1,10)$, to capture varying lengths of term dependencies.

Post TF-IDF vectorization, we use spectral clustering to achieve cluster assignments. We compute clustered EVs for each possible cluster assignment and choose the cluster assignment with the highest variance in EVs across all agents. The computed EVs for the highest-variance cluster assignment are then used as the estimated EVs for all agents.

**Imitation with EV2BC.** We apply EV2Bc by filtering for agents with high estimated EVs. We found that pre-filtering for trajectories with high collective scores significantly improves performance. In other words, we filter for agents with high estimated EVs in trajectories with high collective scores. We assume that this is the case as some agents might 'block' any progress in trajectories with low collective scores.

