# OpenReview forum: "Select to Perfect: Imitating desired behavior from large multi-agent data"
_ICLR.cc/2024/Conference — ICLR 2024 poster_

### Official Review · Reviewer_fLoZ · 2023-10-30

**Soundness:** 2 fair
**Presentation:** 2 fair
**Contribution:** 3 good
**Rating:** 6
**Confidence:** 3

**Summary:**

This paper proposes a method for imitating desired behavior from diverse multi-agent datasets. It introduces the concept of Exchange (EVs), which measures the individual contribution of an agent to a collective value function, and shows how they are related to Shapley Values. It further develops EV-clustering, a technique to estimate EVs from incomplete or anonymized datasets, by maximizing the variance of EVs across clusters of agents. It presents EV-based Behavior Cloning (EV2BC), an imitation learning method that only imitates the actions of agents with high EVs, and demonstrates its effectiveness in two domains: Tragedy of the Commons and Overcooked.

**Strengths:**

- It introduces the concept of Exchange Values, which measure the individual contribution of an agent to a collective value function, and shows how they are related to Shapley Values.
- It develops EV-Clustering, a technique to estimate Exchange Values from incomplete or anonymized datasets, by maximizing the variance of Exchange Values across clusters of agents.
- It presents EV based Behavior Cloning, an imitation learning method that only imitates the actions of agents with high Exchange Values, and demonstrates its effectiveness in two domains: Tragedy of the Commons and Overcooked.

**Weaknesses:**

- For the results, it is necessary to provide some reference for a better understanding of the performance, e.g., reporting the results using the ground truth identification, or the results using the shapely value or other credit assignment methods.
- The paper uses a limited number of environments and datasets to evaluate the proposed method and does not consider more complex or realistic scenarios that involve heterogeneous agents, partial observability, communication, or coordination, e.g., applying the exchange value in proactive multi-camera cooperation[1] or SMAC[2].

Reference:
[1] Ci, Hai, et al. "Proactive Multi-Camera Collaboration For 3D Human Pose Estimation."  ICLR 2023.

[2] Wen, Muning, et al. "Multi-agent reinforcement learning is a sequence modeling problem." NeurIPS 2022.

**Questions:**

- How do you deal with the uncertainty or variability in the EV estimates, especially when the data is incomplete or anonymized? How robust is your method to noise or outliers in the data?
- How do you justify the choice of the DVF for each domain? How do you ensure that the DVF is aligned with the desired behavior and does not have any unintended consequences or biases?

---

> ### Author Response · Authors · 2023-11-20
> **Response to reviewer**
>
> ### Weaknesses
>
> > For the results, it is necessary to provide some reference for a better understanding of the performance, e.g., reporting the results using the ground truth identification, or the results using the shapely value or other credit assignment methods.
>
> Thank you for this remark. We would first like to clarify that, for the human datasets, a ground-truth identification does not exist, as no ground-truth cluster assignments are given for the human participants in this dataset. For the simulated datasets, we can observe in Figure 4 that the EV estimation error is very low if 30% of groups are observed, hence the estimated EVs are very close to ground-truth EVs. The results for EV2BC in Table 3 assume that 30% of groups are observed and hence very closely match the performance using ground-truth identification of agents. We will add the exact results for ground-truth identification, but for now focused on the additional Starcraft experiments.
>
> Unfortunately, it is impossible to compute a Shapley Value baseline in the given scenarios, as Shapley Values cannot be computed for fixed-size games. However, if Shapley Values could be computed (in the absence of group size constraints), then the ordering of agents by Exchange Values is *identical* to the ordering of agents by Shapley Values (see section 4.1 "Relationship between Shapley Value and Exchange Value" for details). Hence, in the absence of group size constraints, using Shapley Values to guide imitation learning will result in identical results as when using Exchange Values. We have also empirically verified this in the Tragedy of Commons environment. We are unaware of other credit-assignment methods that apply to our problem setting but will include a comparison if pointed to such works.
>
> > The paper uses a limited number of environments and datasets to evaluate the proposed method and does not consider more complex or realistic scenarios that involve heterogeneous agents, partial observability, communication, or coordination, e.g., applying the exchange value in proactive multi-camera cooperation[1] or SMAC[2].
>
> We now conducted additional experiments in the Starcraft multi-agent environment, which is a widely used multi-agent coordination benchmark [C, D], which is partially observable, involves long-term planning, requires strong coordination, and where agents are heterogeneous. We find that also here, our method outperforms the baselines by a substantial margin, underlining the applicability of our method to diverse settings. Please consider the general response for details.
>
> ### Questions
>
> > How do you deal with the uncertainty or variability in the EV estimates, especially when the data is incomplete or anonymized? How robust is your method to noise or outliers in the data?
>
> We address the case of noisy or incomplete (individually anonymized) observations by combining both EV-Clustering and behavior clustering. This allows us to achieve good performance even on individually anonymized, noisy, real-world data, as demonstrated by the experiments on the human dataset in the Overcooked environments (see results for $D^{human}$ in Table 1).
>
> > How do you justify the choice of the DVF for each domain? How do you ensure that the DVF is aligned with the desired behavior and does not have any unintended consequences or biases?
>
> We would like to point out that in the discussion of AI Alignment, a distinction between normative and technical alignment is often made [A]. Our work addresses solely the technical aspect of alignment, and it does not try to answer the question of which norms AI systems should align with. Specifically, we do not try to answer the question of what the correct Desired Value function (DVF) for a given problem is. The DVFs that we chose for the different environments are simply inspired by previous literature. A possible option for setting DVFs is to crowdsource rankings of collective outcomes from human annotators [B].
>
> Citations:
> [A] “Artificial Intelligence, Values and Alignment”, Gabriel, Minds and Machines, 2020
> [B] "Human-centred mechanism design with Democratic AI", Koster et al., Nature human behaviour, 2022

---

> ### Author Response · Authors · 2023-11-21
> **Additional questions**
>
> Dear reviewer,
>
> we kindly wanted to ask if we were able to address your concerns. If so, we would like to ask if you would be willing to increase your score, given the positive results in Starcraft.
>
> Best wishes,
> The authors

---

> > ### Author Response · Authors · 2023-11-23
> >
> > Dear reviewer, please respond to our rebuttal, there are less than 5 hours left.

---

### Official Review · Reviewer_wrnL · 2023-10-31

**Soundness:** 3 good
**Presentation:** 3 good
**Contribution:** 3 good
**Rating:** 8
**Confidence:** 2

**Summary:**

This paper addresses the problem of learning aligned imitation policies from multi-agent datasets containing unaligned agents. The authors argue that existing methods for estimating demonstrator expertise in single-agent settings do not translate to the multi-agent setting due to the challenge of credit assignment. This paper proposes a method for learning aligned imitation policies that takes into account the collective value function of the agents. Empirical evidence shows that their proposed method outperforms relevant baselines.

**Strengths:**

Learning human-aligned policies from a mixed multi-agent dataset is an important area of research that is relevant to a diverse set of applications, including autonomous driving. The proposed method takes into account the collective value function of the agents and is designed to address the challenge of credit assignment in multi-agent settings.

This work introduces a new metric called the Exchange Value (EV), which is used to estimate the individual contributions of agents to the collective value function. The paper provides empirical evidence that the proposed method outperforms relevant baselines, by showing that it can be applied to a social dilemma game and a cooperative task.

**Weaknesses:**

The proposed method assumes that the collective value function can be expressed as a sum of individual contributions. The authors should comment more on the class of problems that this is applicable to.

Experiments evaluate the proposed method on a limited set of environments and tasks, and it is unclear how well the method would generalize to other domains and tasks.The authors motivated with a mixed driving dataset, and it would be useful to see how this method applies to driving benchmarks.

**Questions:**

Is EV a good measure if there exist complex/bipolar dynamics between agent behaviors? e.g. two agents work well if they are both in the team but horribly if only one present?

**Details Of Ethics Concerns:**

This paper itself does not comment on potential ethical issues. The notion of exchange value can potentially be applied to evaluate human performance/value to a team/company, which should require extra caution. The idea of evaluating value of an individual's contribution based on how much it differs by swaping them out with a random person can be concerning if applied to human performance.

---

> ### Author Response · Authors · 2023-11-20
> **Response to reviewer**
>
> ### Weaknesses
> > The proposed method assumes that the collective value function can be expressed as a sum of individual contributions. The authors should comment more on the class of problems that this is applicable to.
>
> We would like to point out that this assumption does not apply to Exchange Values or EV2BC, but is merely made to arrive at a theoretical motivation for EV-Clustering.
>
> The inessential game assumption is commonly made when estimating Shapley Values [A], as it makes theoretical analysis tractable. It represents the simplest non-trivial class of cooperative games, and hence a natural first step. Future work could investigate defining the game as a weighted graph (possibly with self-loops), where the value of a group is the total weight of its internal edges (inessential games correspond to graphs where only self-loops have non-zero weight). We now added this clarification to the main paper.
>
> Even when the inessential game assumption is not made, the objective of EV-Clustring, i.e. maximizing the variance in EVs across all agents, is nonetheless well motivated, as this objective results in maximally distinct EVs, while a random partition into clusters would have very low variance (and hence agents would have very similar EVs).
>
> > Experiments evaluate the proposed method on a limited set of environments and tasks, and it is unclear how well the method would generalize to other domains and tasks. The authors motivated with a mixed driving dataset, and it would be useful to see how this method applies to driving benchmarks.
>
> To underline the scalability of our method we now added experiments in the Starcraft multi-agent environment (SMAC, [A]), which we picked as it is probably the most widely adopted benchmark in the multi-agent community [C, D] and has various predefined scenarios and benchmarks (unlike multi-agent driving). It is partially observable, involves long-term planning, requires strong coordination, and is heterogeneous (as also described by Reviewer floZ). We find that also here, our method outperforms the baselines by a substantial margin, underlining the applicability of our method to diverse settings. Please consider the general response for details.
>
> ### Questions
> > Is EV a good measure if there exist complex/bipolar dynamics between agent behaviors? e.g. two agents work well if they are both in the team but horribly if only one present?
>
> Our work makes the assumption that agents behave relatively consistently across trajectories. If some agents A and B perform well when a third agent C is not present, but perform badly when agent C is present (all other things being equal), then this requires splitting the behavior data of agents A and B into two subagents each, which represent agent A and B in the presence and absence of agent C. We highlight this as an interesting direction for future work in our conclusion.
>
> ### Ethics Concern
> > This paper itself does not comment on potential ethical issues. The notion of exchange value can potentially be applied to evaluate human performance/value to a team/company, which should require extra caution. The idea of evaluating value of an individual's contribution based on how much it differs by swaping them out with a random person can be concerning if applied to human performance.
>
> Shapley Values and Exchange Values estimate the alignment of an individual with a group value function (which must be prescribed separately), and as such can be misused if they are included in a larger system that is used to judge those individuals in any way. Discrimination of individuals based on protected attributes is generally unlawful, and care must be taken to avoid any discrimination by automated means. In our work, we demonstrated a novel positive use of these methods by using them to train aligned (beneficial) agents, that do not imitate negative behaviors in a dataset. We expect that the benefits of addressing the problem of unsafe behavior by AI agents outweigh the downsides of misuse of Shapley Values and Exchange Values, which are covered by existing laws. We further clarified this in the Conclusion.
>
> Citations:
> [A] "A unified approach to interpreting model predictions", Lundberg et al, NerurIPS, 2017

---

### Official Review · Reviewer_e3wr · 2023-10-31

**Soundness:** 4 excellent
**Presentation:** 3 good
**Contribution:** 3 good
**Rating:** 8
**Confidence:** 3

**Summary:**

This work proposes a way to imitate the correct agents in a multi-agent setting, where correctness is measured by how an agent impacts the collective. This is termed the exchange value, and is formally presented as a way to quantify an agent’s contribution as the expected change in desirability when substituting the agent randomly. EVs require that different combination of agents are seen in the dataset. To counteract low-data regimes, EV-clustering is proposed. With this exchange values estimated for all agents in a dataset, the authors device EV2BC method learns behaviour cloning policies good agents only. Evaluation is performed on tragedy of commons and on the dataset collected from the Overcooked environment with diverse agent behaviours.

**Strengths:**

- The work is well motivated and seems to solve practical issues in Shapely Values.

- The presentation of the work is clear enough, and relatively easy to follow. I thought Sections 3 and 4 to be quite well written.

- The results seem quite strong and convincing compared to other baseline methods. I thought Figure 4 (right) was quite convincing in showing the importance of clustering for the degenerate case.

**Weaknesses:**

- How scalable would this method be to let's say a dataset of multi-agent driving scenes? It seems to me like scalability is an issue here, specifically due to clustering. this brings me to an important point, the weaknesses of the proposed approach should be addressed.

- I'm still left confused by the differences between behavioural clustering and EV-clustering. I understand the differences in the approaches, but the ablation study seems to point to behavioural clustering being more stable in low-data regiments. I see that the ablation study says to look at section 5.1 to show why behavioural clustering is not sufficient by itself, but I do not see the supporting results.

**Questions:**

- I'm a little confused about some experimental details. Specifically, the number of agents in the datasets, and the exact composition of the dataset is unclear. Can you clarify the composition of the $D^\text{adv}$? Are there really two different types of agents, but $n=100$ agents?

- In my opinion, the paper would be better organized by moving more results from the appendix into the main paper. One simple way of improvement is to move Figure 3 to the appendix and add the ablation study on the EV-clustering vs. Behavior clustering.

---

> ### Author Response · Authors · 2023-11-20
>
> ### Weaknesses:
> > How scalable would this method be to let's say a dataset of multi-agent driving scenes? It seems to me like scalability is an issue here, specifically due to clustering.
>
> To underline the scalability of our method we now added experiments in the Starcraft multi-agent environment (SMAC, [A]), which we picked as it is probably the most widely adopted benchmark in the multi-agent community [C, D] and has various predefined scenarios and benchmarks (unlike multi-agent driving). It is partially observable, involves long-term planning, requires strong coordination, and is heterogeneous (as also described by Reviewer floZ). We find that also here, our method outperforms the baselines by a substantial margin, underlining the applicability of our method to diverse settings. Please consider the general response for details.
>
> > this brings me to an important point, the weaknesses of the proposed approach should be addressed.
>
> We would like to point the reviewer to the conclusion, which already addressed three main limitations of our work. We have now added that we see the application of our method to even larger multi-agent environments, such as driving, as a relevant direction for future research.
>
> > I'm still left confused by the differences between behavior clustering and EV-clustering. I understand the differences in the approaches, but the ablation study seems to point to behavioral clustering being more stable in low-data regiments. I see that the ablation study says to look at section 5.1 to show why behavioral clustering is not sufficient by itself, but I do not see the supporting results.
>
> Thank you for this remark; we now discuss this matter in more detail in Section 5.1 in the main paper. We first recall that behavior clustering only considers low-level behavior data (actions taken by individual agents), while EV-clustering only considers collective outcomes (DVF scores assigned to collective trajectories). In general, EV-clustering should be preferred to behavior clustering, for the following reason. Consider a case where two agents contribute equally to the DVF, in which case both agents should be assigned equal EVs, but both agents do show very different low-level behavior. When applying EV-Clustering, both agents would be assigned to the same cluster and would hence correctly be assigned the same EV. However, when applying behavior clustering, both agents would be assigned to different clusters, which would potentially result in erroneously assigning different EVs to each agent (and would likely result in higher-variance EV estimates overall). EV-clustering may have high variance if only very few collective outcomes are observed for some agents, in which case we can combine it with behavior clustering.
>
> ### Questions:
> > I'm a little confused about some experimental details. Specifically, the number of agents in the datasets, and the exact composition of the dataset is unclear. Can you clarify the composition of the
> $D^{adv}$ Are there really two different types of agents, but $n=100$ agents?
>
> Yes, the $D^{adv}$ dataset contains 100 agents but only two different types of agents (some behaving optimally, some behaving adversarially.) In contrast, the $D^{\lambda}$ dataset contains 100 different agents, where $\lambda$ is sampled from a uniform distribution for each agent. We chose to also have 100 agents in $D^{adv}$, to make it equal in size to $D^{\lambda}$.
>
> > In my opinion, the paper would be better organized by moving more results from the appendix into the main paper. One simple way of improvement is to move Figure 3 to the appendix and add the ablation study on EV-clustering vs. Behavior clustering.
>
> Thank you for this suggestion. We moved Figure 3 to the Appendix. As we also have to accommodate the additional Starcraft experiments in the main paper, we were unable to move Figure 6 to the main paper, but we now discuss the findings from the ablation study, as well as the pros and cons of both clustering methods, in more detail in Section 5.1.
>
> Citations:
> [A] "The Starcraft multi-agent challenge.”, Samvelyan et al., AAMAS, 2019
> [B] “Grandmaster level in StarCraft II using multi-agent reinforcement learning”, Vinayls et al., Nature, 2019

---

> ### Author Response · Authors · 2023-11-21
>
> Dear reviewer, please let us know if we were able to address your concerns. In light of the positive additional experiments in Starcraft, we would also like to ask you to consider raising your score.

---

> > ### Author Response · Authors · 2023-11-23
> >
> > Dear reviewer, please respond to our rebuttal, there are less than 5 hours left.

---

### Official Review · Reviewer_CrLG · 2023-11-01

**Soundness:** 2 fair
**Presentation:** 3 good
**Contribution:** 3 good
**Rating:** 8
**Confidence:** 3

**Summary:**

### Problem Statement
The paper tackles the challenge of extracting and imitating desirable behaviors from large multi-agent datasets, where desirability is quantified via collective trajectory scores. The problem arises as these scores reflect collective outcomes, making it difficult to ascertain individual agents' contributions, especially in real-world scenarios with fixed group sizes, incomplete datasets, and fully anonymized data.

### Main Contributions
The key contributions include
1. The introduction of "Exchange Values" (EVs) to quantify an individual agent's contribution to collective desirability.
2. The proposal of "EV-Clustering" to estimate these values from incomplete datasets
3. The development of "Exchange Value based Behavior Cloning" (EV2BC), a method that selectively imitates agents with high EVs estimated from anonymized data, thus aligning learned policies with desired characteristics, outperforming relevant baselines.

### Methodology
The authors propose "Exchange Values", a modification to Shapley value computation that compares the Desired Value Function values between agent groups of the same size, making it amenable to games that have group size constraints. Based on the Exchange Values, clustering of agents can be done by maximizing inter-cluster EV variance, which is particularly useful for fully-anonymized data. Behavior cloning (BC) can be then confined to only mimicking agents with high Exchange Values.

### Experiments
Two environments are used to evaluate the methods, namely the "Tragedy of Commons" and "Overcooked". Both synthesized and human generated data are used. The experiments show that the estimated EV values are meaningful and superior BC performance is attained with the guidance of EV for selecting trajectories to imitate.

**Strengths:**

### Originality and Significance
Evaluating the agent quality / contribution from desirability scores of collective trajectories is a very realistic and meaningful problem. The proposed method is well-motivated and elegantly extending the well-known Shapley Value, which is innovative.

### Quality
The problems the authors address are important and practical and the questions they try to answer are insightful.

### Writing
The mathematical explanations of complex concepts are precise and consistent. In addition, the authors provide insightful intuition to help readers understand.

**Weaknesses:**

### Limited Environments
Only two environments are studied, while there are many environments that can further highlight the real-world value of the proposed method, e.g. public traffic.

### Lack of more theoretical analysis of the properties of EV
Shapley Values are know to have good properties, e.g. symmetry, dummy (zero value for null players), additivity etc, which make it interpretable, appealing, and useful. It would be interesting to see analysis of Exchange Values with respect to these properties.

### Lack of interpretation of EV
More detailed analysis can be added to the main text with respect to how different EV values can be connected to various behavior patterns. In particular, I think the $\lambda$ values in both "Tragedy of Commons" and "Overcooked" can be linked to the estimated EVs to validate the method.

### Lack of baseline
I understand that this is the first work tackling this specific problem setup, but I'm interested to see whether Shapley values could be similarly useful for guiding imitation learning in multi-agent dataset when the group size constraint is absent (which should be possible in many cases, e.g. the Tragedy of the Commons).

### Writing
Although I in general enjoyed reading the paper, I still find many sentences throughout the article a bit repetitive and convoluted.

**Questions:**

- What does the $m$ in line 220 denote? Is it a fixed value or can it take multiple possible values (since $m \in M$). Why must $k \geq m$? Should $k$ change when $m$ takes a different value?
- Could authors further explain the definition of the "cluster-centroid agents $C \subseteq K$" in line 223?

---

> ### Author Response · Authors · 2023-11-20
> **Response to reviewer part 2**
>
> > Could authors further explain the definition of the "cluster-centroid agents $C \subseteq K$" in line 223?
>
> C represents a group of cluster-centroid agents, with $C \subseteq K$, and where K is the set of all clusters (or set of all cluster-centroid agents). Generally, a cluster-centroid agent can be thought of as a meta-agent that represents the aggregate of all agents assigned to a given cluster, which is done by grouping the observations for all agents assigned to the same cluster into one cluster-centroid agent (a meta-agent). We can then compute outcomes for groups of cluster-centroid agents C, just like we can compute outcomes for groups of individual agents. Assuming adequate cluster assignments, this grouping of observations allows for a significant reduction in the variance in the EV estimates.

---

> ### Author Response · Authors · 2023-11-20
> **Response to reviewer part 1**
>
> ### Weaknesses
>
> > Limited Environments: Only two environments are studied, while there are many environments that can further highlight the real-world value of the proposed method, e.g. public traffic.
>
> To underline the scalability of our method we now added experiments in the Starcraft multi-agent environment (SMAC, [A]), which we picked as it is probably the most widely adopted benchmark in the multi-agent community [C, D] and has various predefined scenarios and benchmarks (unlike multi-agent driving).
> SMAC involves long-term planning, requires strong coordination, and is heterogeneous (as also described by Reviewer floZ). We find that also here, our method outperforms the baselines by a substantial margin, underlining the applicability of our method to diverse settings. Please consider the general response for details.
>
> > Lack of more theoretical analysis of the properties of EV
> Shapley Values are known to have good properties, e.g. symmetry, dummy (zero value for null players), additivity, etc, which make it interpretable, appealing, and useful. It would be interesting to see an analysis of Exchange Values with respect to these properties.
>
> Thank you for this remark. The Shapley Value satisfies four main axioms: Dummy, Efficiency, Symmetry, and Linearity. We have now further clarified the relationship between Shapley Values and Exchange Values in the main paper, and added further clarifications about the different axioms. In Appendix A.1 we provide the following proofs:
> - for the relationship between Shapley Values and Exchange Values
> - that the Exchange Value satisfies the symmetry and linearity axioms
> - that the Exchange Value satisfies appropriately modified versions of the dummy and efficiency axioms.
>
> > Lack of interpretation of EV: More detailed analysis can be added to the main text with respect to how different EV values can be connected to various behavior patterns. In particular, I think the $\lambda$
>  values in both "Tragedy of Commons" and "Overcooked" can be linked to the estimated EVs to validate the method.
>
> Thank you for this remark. Such plots are given in the appendix (due to lack of space), and we now improved their descriptions and referencing in the main paper. Figure 6 shows a plot of the relation of $\lambda$ and EV in the Overcooked environments, and Figure 8 shows the relation between EVs and different strategies in Tragedy of Commons. The strong correlation in Figure 6, as well as the visibly correct ordering of agents in Figure 8, provide insights into the relation between EVs and different strategies. This verifies that EVs are adequate estimators for individual contributions in both environments.
>
> > Lack of baseline: I understand that this is the first work tackling this specific problem setup, but I'm interested to see whether Shapley values could be similarly useful for guiding imitation learning in multi-agent datasets when the group size constraint is absent (which should be possible in many cases, e.g. the Tragedy of the Commons).
>
> Thank you for this remark; we now further clarified this in the main paper. As described and shown in Section 4.1, if Shapley Values can be computed (in the absence of group size constraints), then the ordering of agents by Exchange Values is **identical** to the ordering of agents by Shapley Values. Hence, in the absence of group size constraints, using Shapley Values to guide imitation learning will result in identical results as using Exchange Values. We also empirically verified this in the Tragedy of Commons environment.
>
> > Writing: Although I in general enjoyed reading the paper, I still find many sentences throughout the article a bit repetitive and convoluted.
>
> We appreciate this feedback. We have made several improvements, also following other reviewers' suggestions, and welcome any further suggestions.
>
> ### Questions
> > What does the m in line 220 denote? Is it a fixed value or can it take multiple possible values (since $m \in M$). Why must $k \geq m$? Should k change when m takes a different value?
>
> We apologize for the misunderstanding. This is a typo that we have now fixed. It now reads $k \leq n$, where n is the number of agents and k is the number of clusters.

---

> > ### Comment · Reviewer_CrLG · 2023-11-21
> >
> > I thank the authors for their response, which addresses most of my questions. However I'm still struggling to understand the EV-clustering: Is it the agents that are being clustered? If so, with $k \geq n$, is it true that there are more clusters than agents to assign?

---

> > > ### Author Response · Authors · 2023-11-21
> > >
> > > Dear reviewer,
> > >
> > > you are absolutely right. It is $k \leq n $, i.e., the number of clusters must be smaller or equal to the number of agents. In practice, the number of clusters is much smaller than the number of agents (for example, for the additional Starcraft experiments, we assigned 200 agents to 5 clusters). Thank you for spotting this, we have fixed it now.
> > >
> > > Does this (and the additional explanation in the second part of our response) answer your questions about EV-clustering?

---

> ### Author Response · Authors · 2023-11-21
>
> You pointed out that we were able to address most of your questions by providing the proofs asked for and running the additional experiments in Starcraft. We kindly wanted to ask whether you would be willing to increase your score?

---

> ### Comment · Reviewer_CrLG · 2023-11-23
>
> Thank you for the response. The added StarCraft experiments is encouraging, and the enriched theoretical analysis is valuable.
> I would have raised my rating to 7, but since 7 is not a choice, I keep my rating of 6.
>
> I must point out that there still exist minor errors in the manuscript. To name a few,
>
> 1. The "Table 3" in line 367 is referring to "Figure 3", which is actually a table
> 2. The caption of Figure 6 describes that "The brighter, the higher an agent’s contribution to a given value function", but there does not seem to be any brightness variation in the plot.
>
> Also, please provide more explanation on the various settings in the StarCraft task.

---

> > ### Author Response · Authors · 2023-11-23
> >
> > Dear reviewer,
> >
> > Thank you for this feedback, we will update this immediately.
> >
> > Please consider raising your score to 8 (as indicated in your original comment), as you this would make an important change, given that you are the only reviewer that engaged with our rebuttal.
> >
> > Best wishes,
> > The authors

---

> > > ### Author Response · Authors · 2023-11-23
> > >
> > > Dear reviewer, we have fixed both errors and will add a detailed description of the Starcraft experiments to the appendix.

---

### Official Review · Reviewer_VYWJ · 2023-11-01

**Soundness:** 3 good
**Presentation:** 2 fair
**Contribution:** 2 fair
**Rating:** 5
**Confidence:** 3

**Summary:**

The paper "Who to Imitate: Imitating Desired Behavior from Diverse Multi-Agent Datasets" proposes a novel imitation learning framework. It enables AI agents to learn desirable behaviors from large, mixed-quality multi-agent datasets by using a metric called Exchange Value (EV) to evaluate and imitate only those agents contributing positively to collective outcomes. The technique involves EV-Clustering to handle incomplete data and an Exchange Value based Behavior Cloning (EV2BC) method for learning policies aligned with desired outcomes. The approach is shown to outperform baselines and has applications in aligning AI behavior with human values in complex environments.

**Strengths:**

Innovative Metric for Agent Evaluation: The introduction of Exchange Values (EVs) as a metric to compute individual agents' contributions to a collective value function is a significant contribution. EVs offer a method for identifying and imitating desirable behaviors within multi-agent systems, providing a novel way to approach imitation learning.

Effective Handling of Incomplete Data: The paper presents EV-Clustering, a method that estimates contributions from incomplete datasets. This addresses a common challenge in real-world scenarios where datasets are rarely comprehensive, enabling more accurate modeling of agent behavior.

Alignment with Desirable Outcomes: Through Exchange Value based Behavior Cloning (EV2BC), the paper proposes a mechanism to align the learning process with a Desired Value Function (DVF). This ensures that the learned policies reflect desirable outcomes, which is crucial for the practical application of AI systems trained on human data.

**Weaknesses:**

1. **Quantification of Desirability**: The process of quantifying the desired value function (DVF) is a complex task and an active area of research. The paper's methods depend on the DVF to guide the imitation learning process, so any limitations in accurately defining this function could impact the effectiveness of the approach.

2. **Assumption of Consistent Agent Behavior**: The framework assumes that individual agents behave similarly across multiple trajectories. This assumption may not always hold true in complex, dynamic environments where agent behavior can vary significantly based on context.

3. **Utilization of Undesired Behavior Data**: The paper points out that further research could explore how to utilize data on undesired behavior more effectively, such as developing policies that are explicitly constrained to avoid undesirable actions.

**Questions:**

1. Given that the quantification of what is considered desirable behavior is central to the proposed framework, can the authors provide additional insights into how the Desired Value Function (DVF) is defined and quantified across different environments and datasets?

2. The paper assumes consistent behavior from individual agents across multiple trajectories. Could the authors discuss the potential implications of this assumption in environments where agent behavior is more dynamic and context-dependent?

3. The paper suggests the potential for utilizing data on undesired behavior more effectively. Could the authors elaborate on possible approaches for leveraging this type of data to enhance the imitation learning process?

4. How does the framework adapt to different environments, and what are the limitations when applying the proposed EV-Clustering and EV2BC methods to datasets that significantly differ from the ones used in the experiments?

---

> ### Author Response · Authors · 2023-11-20
> **Rebuttal response**
>
> ### Questions
> > Given that the quantification of what is considered desirable behavior is central to the proposed framework, can the authors provide additional insights into how the Desired Value Function (DVF) is defined and quantified across different environments and datasets?
>
> We would like to point out that in the discussion of AI Alignment, a distinction between normative and technical alignment is often made [A]. Our work addresses solely the technical aspect of alignment, and it does not try to answer the question of which norms AI systems should align with. Specifically, we do not try to answer the question of what the correct Desired Value Function (DVF) for a given problem is. The example DVFs that we chose for the different environments are simply inspired by previous works and intuition. A possible option for setting DVFs is to crowdsource rankings of outcomes from human annotators [E].
>
> > The paper assumes consistent behavior from individual agents across multiple trajectories. Could the authors discuss the potential implications of this assumption in environments where agent behavior is more dynamic and context-dependent?
>
> Our method can be extended to situations where agents’ behavior differs across multiple trajectories by splitting up the data of an individual agent into different sub-agents and treating each sub-agent as its own agent. This extension of our work does not require additional methodological contributions, and we see it as an interesting direction for future work, as also outlined in our conclusion.
>
> > The paper suggests the potential for utilizing data on undesired behavior more effectively. Could the authors elaborate on possible approaches for leveraging this type of data to enhance the imitation learning process?
>
> Generally, data on undesired behavior could be used to disincentivize the policy from imitating undesired behavior or to gain further understanding of the environment dynamics. While some work exists in this field [B], this constitutes a highly interesting direction for future work.
>
> > How does the framework adapt to different environments, and what are the limitations when applying the proposed EV-Clustering and EV2BC methods to datasets that significantly differ from the ones used in the experiments?
>
> We are not entirely sure we understand this question correctly, and kindly ask for potential clarification. Our framework applies to environments that can be modeled as multi-agent MDPs, and where a function is given that assigns values to collective outcomes. To further showcase the applicability of our approach, we have now conducted additional experiments in the Starcraft multi-agent environment (SMAC, [C]). SMAC is a widely used multi-agent coordination benchmark [C, D], which is partially observable, involves long-term planning, requires strong coordination, and is heterogeneous (as also described by Reviewer floZ). We find that also here, our method outperforms the baselines by a substantial margin, underlining the applicability of our method to diverse settings. Please consider the general response for details.
>
> Citations:
> [A] “Artificial Intelligence, Values and Alignment”, Gabriel, Minds and Machines, 2020
> [B] “Model-based Offline Imitation Learning with Non-expert Data”, Park et al., arXiv, 2022
> [C] "The Starcraft multi-agent challenge.”, Samvelyan et al., AAMAS 2019
> [D] “Grandmaster level in StarCraft II using multi-agent reinforcement learning”, Vinayls et al., Nature 2019
> [E] "Human-centred mechanism design with Democratic AI", Koster et al., Nature human behavior 2022

---

> > ### Comment · Reviewer_VYWJ · 2023-12-04
> >
> > Appreciate the feedback. I've gone through the other reviews and the authors' rebuttal. After evaluating the paper's overall quality, I've decided to maintain my score.

---

> ### Author Response · Authors · 2023-11-21
>
> Dear reviewer, please let us know if we were able to address your concerns. Best wishes, the authors.

---

> > ### Author Response · Authors · 2023-11-23
> >
> > Dear reviewer, please respond to our rebuttal, there are less than 5 hours left.

---

### Author Response · Authors · 2023-11-20
**General Response to all Reviewers**

Dear Reviewers,

thank you for your reviews of our work. We are pleased you found our work well-motivated, relevant to a diverse set of applications, and
an elegant extension of Shapley Values. It seemed that most questions were about the relation between Shapley Values and Exchange Values (including if and how properties of Shapley Values apply to Exchange Values) and about the applicability of our method to larger and more complex settings. We address these below and have updated the pdf accordingly.

### Shapley Values, Exchange Values, Properties:
As described in Section 4.1, if Shapley Values can be computed (in the absence of group size constraints), then the ordering of agents by Exchange Values is **identical to the ordering of agents by Shapley Values**. Hence, in the absence of group size constraints, using Shapley Values to guide imitation learning will result in identical results as using Exchange Values. For the exact relationship between Exchange Values and Shapley Values please refer to Proposition A.1 in Appendix A.1.

The Shapley Value satisfies four main axioms: Dummy, Efficiency, Symmetry, and Linearity. We have now further clarified the relationship between Shapley Values and Exchange Values in the main paper, and added further clarifications about the different axioms. In Appendix A.1 **we provide the following proofs:**
- that the Exchange Value satisfies the symmetry and linearity axioms
- that the Exchange Value satisfies appropriately modified versions of the dummy and efficiency axioms.

### Experiments in the Starcraft multi-agent domain:
The StarCraft multi-agent environment [A, B] is probably the most widely used multi-agent learning benchmark, featuring different predefined scenarios and benchmarks. It is partially observable, involves long-term planning, requires strong coordination, and is heterogeneous (as also described by Reviewer floZ). Observation spaces and action spaces are higher dimensional than in previous experiments, with around 100 and 15 dimensions respectively. We consider the scenarios 2s3z, 3s_vs_5z, and 6h_vs_8z, which involve teams of 3-6 agents. For each scenario, we generate a pool of 200 agents with varying capabilities by extracting policies at different epochs, and from training with different seeds. We generate a dataset that contains simulated trajectories of 100 randomly sampled groups (out of about $10^9$ possible groups) and use the environment's ground truth reward function to assign DVF scores according to the collective performance of agents. We first apply EV-Clustering (combined with behavior clustering by agents' actions), then compute EVs for all agents in the dataset, and lastly, use EV2BC to only imitate the behavior of agents that is aligned with the DVF (behavior of agents which make a large contribution to group performance). We compare the performance achieved by the EV2BC agents to that of agents trained either with standard BC or with Group-BC (which imitates the behavior of agents who are part of groups with high collective DVF scores).

We observe in the table below that EV2BC outperforms the baselines by a substantial margin, underlining the applicability of our method to larger and more complex settings.

Average test-time return for three different scenarios for different methods:
| Imitation method | 2s3z | 3s_vs_5z | 6h_vs_8z |
|-|-|-|-|
| BC              | 13.24 ± 1.26     | 12.73 ± 3.25      |  9.56  ± 0.67   |
| Group-BC        | 17.25 ± 2.05     | 12.32 ± 1.92      |  10.08 ± 1.07   |
| EV2BC (ours)    | **19.46** ± 2.98 | **17.15** ± 2.13  |  **12.25** ± 1.55   |

Note that the OMAR baseline, which is implemented as offline MARL with the DVF as the final-timestep reward, did substantially worse than BC.

Citations:
[A] "The Starcraft multi-agent challenge.”, Samvelyan et al., AAMAS 2019
[B] “Grandmaster level in StarCraft II using multi-agent reinforcement learning”, Vinayls et al., Nature 2019

---

### Author Response · Authors · 2023-11-22
**Please respond to our rebuttal -- less than 20 hours left**

Dear reviewers,

We addressed your questions in detail, we supplied additional proofs, and we ran additional experiments that were requested by three reviewers.

We would highly value your response to our rebuttal and would appreciate it if you would consider updating your scores accordingly.

Thank you for your time.

Best wishes,
The authors

---

### Meta-Review · Area_Chair_MCPa · 2023-12-06

**Metareview:**

This paper proposes a method for multi-agent imitation learning, where the goal is to train agents on diverse datasets (where agents in the dataset could have varying quality). The authors propose to use "exchange value (EV)" to measure an agent's contribution to the collective outcome (the expected change in value if the agent is substituted with a random agent). The authors further show how EV-clustering can be used to estimate EV in settings with few observations, as well as use exchange value based behavior cloning (EV2BC) to train the model to imitate agents with positive contributions. The paper is evaluated on Tragedy of the Commons and Overcooked datasets, with StarCraft being added during the rebuttal. Reviewers appreciate the conceptual contribution of EV as well as the promising experimental results. The additional StarCraft experiment also works to demonstrate the model at larger scales (3~6 agents, with a pool of 200 agents), although some concerns remain on scalability to real-world datasets. Another limitation is that EV models individual contribution of agents, and would not address more complex cases where agent behavior may vary due to the presence/absence of other agents. Regardless, I recommend acceptance of this work due to the conceptual contribution and associated experiments. One note is that the newly added StarCraft experiments are not as carefully documented as the original datasets, and the authors are encouraged to update the paper to include more details for reproducibility.

Note to authors: there is a typo in the title (divserse --> diverse)

**Justification For Why Not Higher Score:**

While the work is promising for multi-agent systems, there remains concerns on the scalability of the method to real-world systems. Additionally, there are still some limitations of the EV metric in modeling complex, inter-dependent agent interactions (whereas EV measures individual agent contribution), as outlined by some reviewers.

**Justification For Why Not Lower Score:**

Reviewers agree on the technical contributions of the paper for multi-agent imitation learning, and the author's proposed exchange value metric for measuring the contribution of an agent to the collective. The author's rebuttal also addressed many of the reviewers' concerns - in particular, the StarCraft experiments were helpful as an additional domain.

---

### Decision · Program_Chairs · 2024-01-16

Accept (poster)